# Contribution of warm and moist atmospheric flow to a record minimum July sea ice extent of the Arctic in 2020.

Yu Liang[1,2,4], Haibo Bi[1,2,3], Haijun Huang[1,2], Ruibo Lei[4,7], Xi Liang[5], Bin Cheng[6], Yunhe Wang[1]

[1] Key Laboratory of Marine Geology and Environment, Institute of Oceanology, Chinese Academy of Sciences, Qingdao, 266071, China

[2] University of Chinese Academy of Sciences, Yuquan Road 19, Beijing 100049, China

[3] Laboratory for Marine Geology, Qingdao National Laboratory for Marine Science and Technology, Qingdao, 266061, China

[4] Key Laboratory for Polar Science, MNR, Polar Research Institute of China, Shanghai 200136, China

[5] Key Laboratory of Research on Marine Hazard Forecasting Center, National Marine Environmental Forecasting Center, Beijing, 100081, China

[6] Polar Meteorology and Climatology, Finnish Meteorological Institute, Helsinki, PO Box 33 FIN-00931, Finland

[7] Technology and Equipment Engineering Centre for Polar Observations, Zhejiang University, Zhoushan 316000, China

*Correspondence to*: Haijun Huang (hjhuang@qdio.ac.cn); Haibo Bi (bhb@qdio.ac.cn)

**Abstract.** The satellite observations unveiled that the July sea ice extent of the Arctic shrank to the lowest value in 2020 since 1979, with a major ice retreat in the Eurasian shelf seas including Kara, Laptev, and East Siberian Seas. Based on the ERA-5 reanalysis products, we explored the impacts of warm and moist air-mass transport on this extreme event. The results reveal that anomalously high energy and moisture converged into these regions in the spring months (April to June) of 2020, leading to a burst of high moisture content and warming within the atmospheric column. The convergence is accompanied by local enhanced downward longwave surface radiation and turbulent fluxes, which is favorable for initiating an early melt onset in the region with severe ice loss. Once the melt begins, solar radiation played a decisive role in leading to further sea ice depletion due to ice-albedo positive feedback. The typical trajectories of the synoptic cyclones that occurred on the Eurasian side in spring 2020 agree well with the path of atmospheric flow. Assessments suggest that variations in characteristics of the spring cyclones are conducive to the severe melt of sea ice. We argue that large-scale atmospheric circulation and synoptic cyclones act in concert to trigger the exceptional poleward transport of total energy and moisture from April to June to cause this record minimum of sea ice extent in the following July.

## 1 Introduction

Arctic sea ice is declining dramatically (Wang et al., 2019) under the background of global warming (Comiso and Hall, 2014) and Arctic Amplification (Serreze and Barry, 2011; Kim et al., 2016). Besides, the environment of the Arctic has transformed to a new state with younger (Rigor and Wallace, 2004; Tschudi et al., 2016) and thinner (Kwok and Rothrock,

2009; Bi et al., 2018) ice floes. The severe retreat of Arctic sea ice provides vital implications of environmental change, with a diverse impact on regional and even global climate (Overland et al., 2015; Gu et al., 2018; Previdi et al., 2020), marine ecology (Post et al., 2013), economic activities (Crépin et al., 2017). Likewise, the scientific studies about the causes of Arctic sea ice shrinkage encompass various disciplines, including atmospheric (Deser et al., 2000; Wu et al., 2006; Wang et al., 2009; Ogi et al., 2016) and oceanic (Årthun et al., 2012; Miles et al., 2014; Zhang, 2015; Årthun and Eldevik, 2016) sciences.

Regarding the sea ice-atmosphere interactions, previous studies provided a consensus that changes in both local large-scale atmospheric circulation (Wu et al., 2006; Deser and Teng, 2008; Ding et al., 2017; Lei et al., 2019; Bi et al., 2021) and synoptic activities (e.g., cyclones) (Zhang et al., 2013a; Olason and Notz, 2015; Wernli and Papritz, 2018; Lei et al., 2020) could significantly impact sea ice variation in the Arctic. Some studies also linked the sea ice loss to large-scale circulation changes and forcing from the tropics through teleconnections (Baxter et al., 2019; Screen and Deser, 2019; Warner et al., 2020). Changes in the atmospheric pattern play an important role in regulating the variations and trends in sea ice through different mechanisms. With respect to the thermodynamics, the heat and moisture advection from mid-latitudes increase the air temperature, humidity, and cloudiness, thereby altering the surface radiation and energy budget in the Arctic (Doyle et al., 2011; Graversen et al., 2011; Dufour et al., 2016; Vázquez et al., 2017; Papritz, 2020). The enhanced downward surface longwave radiation associated with heat and moisture transport favors an earlier melt onset, which regulates the surface energy uptake throughout the melt season (Mortin et al., 2016; Kapsch et al., 2019; Horvath et al., 2021). Dynamically, wind anomalies associated with atmospheric conditions can induce sea ice motion and redistribution (Brümmer et al., 2001; Olason and Notz, 2015; Liang et al., 2021), which can affect the sea ice transport through Fram Strait, the Baffin Bay, and channels in the Canadian Arctic Archipelago (Kwok, 2006; Smedsrud et al., 2017; Bi et al., 2019). Besides, inhomogeneous wind anomalies can deform the sea ice cover and cause cracks, leads, and polynyas (Lei et al., 2020).

Sea ice extent (SIE) minima of the Arctic reached its record low in September 2012 during the period 1979-2020, stood at $3.39 \times 10^6$ km$^2$. After that, 2020 witnessed the second-lowest September SIE in the Arctic, which ended up with $3.82 \times 10^6$ km$^2$. During the seasonal cycle of the sea ice cover in 2020, SIE grew to its maximum on March 5, then decreased persistently in the following warm months. Roughly speaking, Arctic sea ice cover was smaller in extent during spring and early summer than that of 2012. Consequently, in 2020, Arctic sea ice experienced the lowest July extent recorded since 1979 (Fig. 1a). As estimated, the July SIE of 2020 shrunk to $7.29 \times 10^6$ km$^2$, which is ~5% (or ~20%) lower than that of 2012 (or the 1979-2020 average July SIE). Figure 1b demonstrates the spatial pattern of sea ice concentration (SIC) anomalies and the corresponding SIE in July 2020. During the early summer of 2020, a prominent SIC reduction occurred in the Eurasian shelf seas, including Kara, Laptev, and East Siberian Seas (60° E-165° E and 70° N-82° N, purple polygon in Fig. 1, hereafter the study area). The averaged SIC anomaly in these areas of July 2020 (-25.96%) exceeds corresponding anomalies

from all other years during the period 1979-2020. Therefore, the sea ice retreat in these regions contributed remarkably to the distinguished shrinkage of SIE in the Arctic Ocean in July 2020.

At present, the record low Arctic sea ice extent (SIE) in July 2020, especially in the study area (Fig. 1b and c), has not garnered much attention. The underlying mechanisms contributing to this extreme event remain unclear. Inspired by the previous works, we hypothesize that this extreme event is closely related to the atmospheric forcing. We conduct an assessment for the preconditions in meteorological fields during the spring months prior to the record minimum SIE in July 2020. Specifically, we examine the magnitude and variations of the poleward atmospheric transport of total energy and moisture as well as their convergence over the Arctic, especially in the study region, during the spring months (April-June) of 2020. The changes in terms of the temperature and specific humidity fields over the vast area with significant sea ice retreat due to the convergence of the energy and water vapor are explored. To quantify the sea ice melt due to changes in the surface energy balance, the surface energy flux components including downward longwave radiation, solar radiation, and turbulent fluxes are analyzed. Moreover, we investigate the distinct role of the synoptic activities, which contributes to the significant anomalies of the moisture and energy fluxes into the region with substantial ice loss.

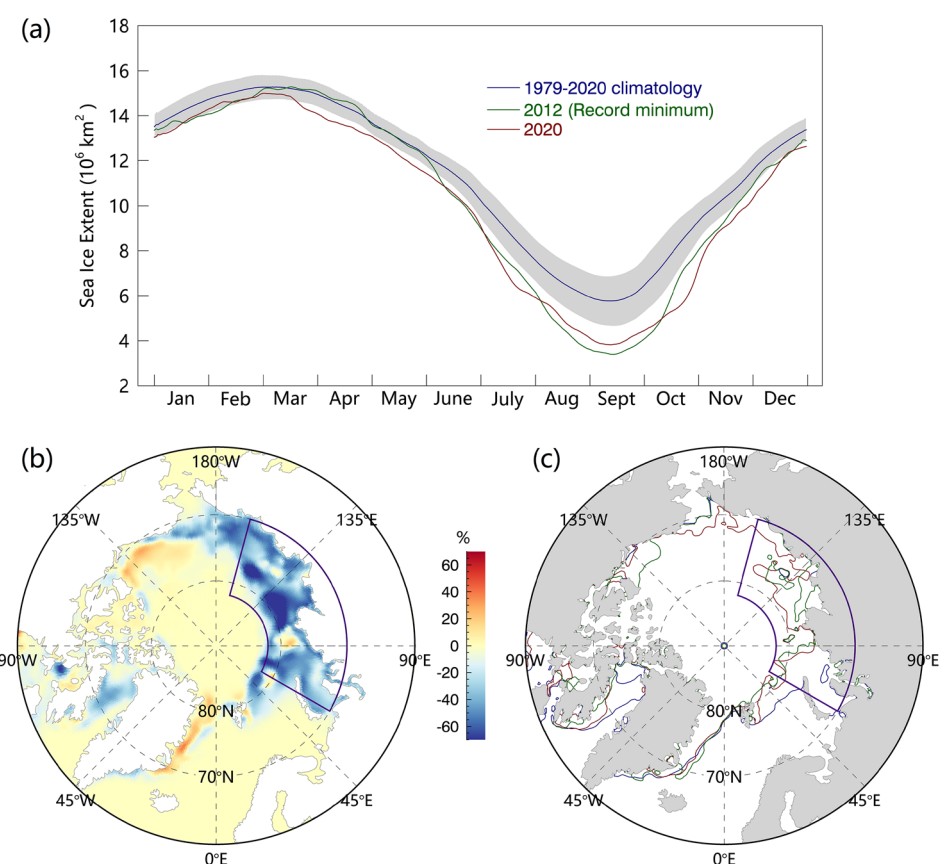

**Figure 1.** (a) Daily SIE of the Arctic in 2020, 2012 and the climatology during 1979-2020. The shadows denote mean minus/plus one standard deviation. (b) Spatial patterns of SIC anomalies (shading), and (c) the SIEs in typical years (bold lines). The red line represents the SIE in July 2020. Green and navy-blue lines denote the SIE in July 2012 and the 42-yr average of the period 1979-2020, respectively. The anomalies are computed as the difference between the fields in July and the corresponding climatology over the past four decades (1979-

2020). Purple polygons encapsulate areas where substantial sea ice cover loss (60° E-165° E, 70° N-82° N) was observed in July 2020, which represents the study area of this paper.

## 2 Data and methods

### 2.1 Data

We use the satellite-derived daily sea ice concentration (SIC) and the Polar Pathfinder Daily sea ice motion (SIM) vectors product provided by the National Snow and Ice Data Center (NSIDC) to investigate the SIE variation and its association with ice advection over the ice-retreated area. SIC fields are available on a polar stereographic projection and are derived from the SMMR, the SSM/I, and the SSMIS by applying the bootstrap algorithm (Comiso, 2017). The latest version (Version 3.1) of the dataset provides improved consistency between sensors through the use of a suite of daily varying tie

points generated from the AMSR-E observations. The SIM product is derived from a variety of sensors on satellite platforms, merged with buoy observations as well as reanalyzed wind data. The motion data is georeferenced to the Equal-Area Scalable Earth (EASE) Grids. The upgrade of the most recent SIM (Version 4.0) addresses artifacts resulting from interpolation, which is considered as one of the most comprehensive sea ice motion datasets (Tschudi et al., 2019). Both datasets have a spatial resolution of 25 km and a temporal resolution of 1-day.

We use daily sea ice thickness (SIT) from the Pan-Arctic Ice Ocean Modeling and Assimilation System (PIOMAS) (Zhang and Rothrock, 2003). PIOMAS is a coupled sea-ice/ocean model forced by atmospheric fields and sea surface temperatures from the National Centers for Environmental Prediction/National Center for Atmospheric Research ((NCEP/NCAR) reanalysis. Multiple studies have compared PIOMAS SIT fields with satellite, submarine, airborne, and in situ observations, whose results revealed that PIOMAS is highly consistent with these observations (Zhang and Rothrock,

2003; Schweiger et al., 2011; Stroeve et al., 2014; Wang et al., 2016). In particular, Schweiger et al. (2011) outlined a less than 0.1-m mean difference and a high pattern correlation (r = +0.8) between the PIOMAS and the ICESat-derived SIT fields. As for the study area, ICESat and PIOMAS ice thickness fields show a close agreement with the spatial distribution o in the Laptev and East Siberian Seas, and the PIOMAS thickness fields are about 0.2-0.7 m thinner than that of ICESat for February–March (Schweiger et al., 2011). The PIOMAS SIT dataset is available on a generalized orthogonal curvilinear

coordinate system with a mean resolution of 22 km.

Cryosphere Science Research Portal (CSRP) of the National Aeronautics and Space Administration (NASA) provides the record of sea ice surface melt dates in the Arctic, from which we obtain the knowledge concerning the onset of the melt season. The fields are derived from SSM/I data following Markus et al. (2009) and available from 1979 to 2020 on the data grid in line with the SIC fields provided by NSIDC.

ERA5 reanalysis datasets including sea level pressure (SLP), temperature, specific humidity, surface evaporation, wind speed, the vertical integral of northward/eastward water vapor flux, the vertical integral of northward/eastward total energy

flux and their convergence, as well as the radiation parameters (surface net/downward longwave radiation, surface net/downward shortwave radiation, surface latent heat flux and surface sensible heat flux) are obtained from the European Centre for Medium-Range Weather Forecasts (ECMWF) (Hersbach et al., 2020). ERA5 represents a new reanalysis product that improves on its predecessor ERA-Interim, which is found to be the most credible reanalysis for the Arctic climate (Kapsch et al., 2013). Compared with ERA-Interim, the major strength of ERA5 is the much higher temporal and spatial resolutions than those of previous global reanalyses and better performance in the troposphere (Hersbach et al., 2020). The adopted ERA5 datasets are characterized by a spatial resolution of 1.0° ×1.0° in longitude and latitude. Note that most of the ERA5 variables we utilized are recorded every six hours except for parameters that accumulated over a particular time period (evaporation and surface radiation fluxes) and the convergence of the energy and moisture transport.

## 2.2 Methods

### 2.2.1 The atmospheric transport of total energy and moisture

The net atmospheric moisture transport for the Arctic represents regionally integrated precipitation minus evapotranspiration and the precipitable water tendency. However, large errors and sparse uncertainties exist in precipitation and evapotranspiration measurements (Zhang et al., 2013b). To this end, we utilize wind and specific humidity fields from reanalysis, which are of great fidelity, to compute moisture transport. The vertical integral of northward moisture flux can be approximated following the trapezoidal rule (Dufour et al., 2016):

$$\int_0^{p_s} f(p)dp \approx \sum_{n=n_0}^{N-1} \frac{1}{2}[f(p_{n+1}) + f(p_n)](p_n - p_{n+1}) + f(p_{n_0})(p_s - p_{n_0}) \tag{1}$$

$$f(p_n) = v_n q_n \tag{2}$$

where $n$ is the number of pressure levels, $n_0$ is the bottom pressure level, $p_s$ denotes the surface pressure. $p_n$ corresponds to pressure at the $n$th pressure level, $v_n$ and $q_n$ represents the northward component of the wind speed and specific humidity at the $n$th pressure level, respectively. We compared our estimated results of the vertical integrated northward water vapor flux against the ERA5 data. The results (not shown) show that the estimated results are highly consistent with the corresponding ERA5 filed both in the magnitude and change of all months across various latitudes (e.g., 70° N) during 1979-2020, which lends credence to the direct use of the water vapor flux field obtained from ERA5. Similarly, we take advantage of the vertical integral of total energy from ERA5. The vertically integrated, atmospheric, northward energy transport consists of internal, potential, kinetic and latent energy.

## 2.2.2 Changes in sea ice thickness due to melt

Changes in surface energy budget related to energy and water vapor convergence could affect sea-ice melting. The sea ice thickness of melt caused by alteration of surface energy budget can be calculated via the sea-ice growth model (Parkinson and Washington, 1979):

$$- \triangle h = \frac{\triangle t}{\rho L}[H_\downarrow + LE_\downarrow + \varepsilon_w LW_\downarrow + (1 - \alpha_w)SW_\downarrow + F_{l\uparrow} + F_{w\uparrow}] \qquad (3)$$

where $\triangle h$ represents sea-ice growth, $\triangle t$ represents the time step, $\rho$ is the density of sea ice (917 kg/m³), $L$ is the latent heat of fusion for sea ice (333.4 kJ/kg), $H_\downarrow$ represents sensible heat, $LE_\downarrow$ represents latent heat, $LW_\downarrow$ corresponds to incoming longwave radiation with $\varepsilon_w$ longwave emissivity, $SW_\downarrow$ corresponds to incoming shortwave radiation with the surface albedo $(\alpha_w)$, $F_{l\uparrow}$ denotes the heat flux, and $F_{w\uparrow}$ denotes the conductive heat flux at the ice-ocean interface. In summer, the conductive heat flux at the ice surface is negligible because the vertical temperature gradient through the ice is close to zero at that time. Here, to focus on the effect of atmospheric forcing on sea ice, we also neglect the heat flux emitted from the surface.    Thereby, the changes in sea ice thickness related to surface atmospheric forcing can be estimated as:

$$-\Delta h = \frac{\Delta t}{\rho L}[FL_{w\downarrow} + FS_{w\downarrow} + H_\downarrow + LE_\downarrow] \qquad (4)$$

where $FL_{w\downarrow}$ and $FS_{w\downarrow}$ represent the surface net fluxes of longwave and shortwave radiation, respectively.

## 2.2.3 Cyclone identification and tracking

To examine the effects of cyclone activities on the anomalous energy and moisture transport in spring 2020, we use a revised automatic cyclone identification and tracking algorithm developed originally by Serreze et al. (1993) to diagnose the center positions and trajectories of the cyclones from the 6-hourly SLP data (Serreze et al., 1993; Wang et al., 2013). The cyclone detection and tracking algorithm consists of two steps: (1) Inspecting each candidate center with a minimum pressure value over the surrounding 7×7 array of grid points. The pressure gradient between the central grid and the surrounding grids should be higher than the detection threshold (0.1 hPa) (Wang et al., 2013). (2) Tracking the centers between two consecutive time steps based on the "nearest neighbor" rule to form trajectories, with further checks including the distance moved in specific directions and pressure tendency. Therefore, a cyclone track consists of a series of cyclone centers identified in sequential time steps at adjacent locations. In this study, thresholds related to multiple parameters, including the maximum travel distance (800 km), maximum north-, south- and west-ward migration (600 km), and maximum pressure tendency (20 hPa) (Serreze, 1995; Wang et al., 2006), are adopted. Note that regions with surface elevations higher than 1000 m are excluded since the algorithm tends to detect spurious systems due to larger uncertainty in the SLP over mountainous terrain.

Moreover, the corresponding features for each cyclone, including the duration, intensity, and radius, were also retrieved. The intensity is referred to as the difference between the SLP of the cyclone center and the climatological monthly mean SLP at corresponding grid points. The density of tracks denotes the number of distinct cyclones occurring in a particular region during spring. We use an integrative parameter, the Cyclone Activity Index (CAI), to measure the intensity, number, and duration of a cyclone. The CAI is defined as the sum of the intensity over all cyclone centers in a particular region during the spring (Zhang et al., 2004). A more detailed description of the automated cyclone detection and tracking scheme can be found in Liang et al. (2021). Neu et al. (2013) conducted an intercomparison experiment involving 15 commonly used detection and tracking algorithms for extratropical cyclones, in which they found that cyclone characteristics are robust between different schemes. Besides, their results revealed that the algorithm used in this study agrees well with the others in terms of spatial distribution, interannual variability, and geographical linear patterns of the cyclones. To some extent, these facts give credence to the method utilized in this study.

## 3 Atmospheric energy and moisture transport

The transport of total energy and moisture toward the Arctic system is controlled by changes in large-scale atmospheric circulation and patterns of climatic variability (Graversen et al., 2011; Vihma et al., 2016). As depicted in Fig. 2, the average Arctic atmospheric condition from April to June 2020 was dominated by a broad and persistent low-pressure anomaly with multi-centers over the central Arctic and extended southwards from the Barents-Kara Seas to the middle part of northern Eurasia. Two high-pressure anomaly centers were located in Eastern Siberia and around the Norwegian Sea, respectively. The low-pressure center in the Barents and two high-pressure centers are quite intensive, with surface pressure values being 1.5 standard deviations below (above) the average value over the past four decades while the low-pressure center near the pole has a relatively smaller magnitude (exceed one standard deviation). These SLP modes favor anomalous southerly winds, which transport moist and warm air mass from Eurasia into the Arctic through entry in the Kara Sea. After entering the Arctic Ocean, the air mass was deflected to move along the coast of Eurasia and influenced the shelf seas. Horvath et al. (2021) pointed out that this kind of atmospheric condition could effectively promote heat and moisture transport into the Arctic. Besides, Kapsch et al. (2019) showed negative geopotential height anomalies over the Arctic favor cyclonic flow from Siberia and the Kara Sea into the eastern Arctic Ocean. Although similarities exist, there are discrepancies in the atmospheric conditions of spring 2020, leading to a unique pattern of moisture and total energy transport.

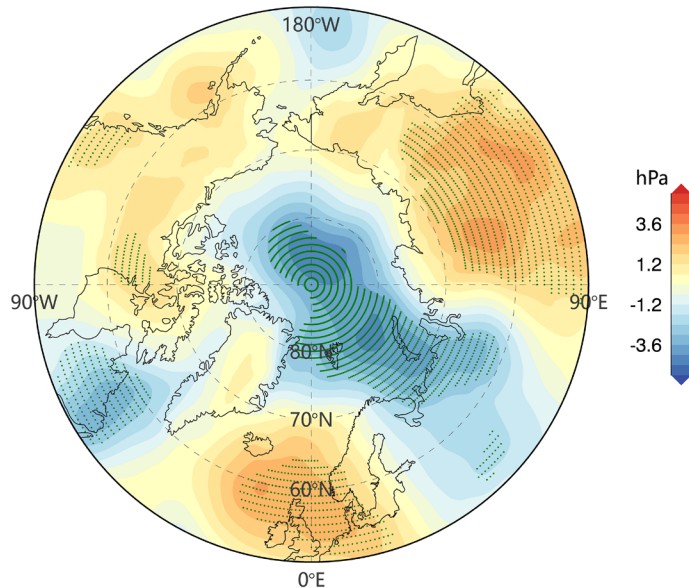

**Figure 2.** Spatial patterns of SLP anomalies during April to June 2020. The anomalies are computed as the difference between the averaged fields of the three months (April-June) and the corresponding climatology over the past four decades (1979-2020). Stipplings represent the values where the anomaly exceeds one standard deviation.

Using ERA5 reanalysis, we quantify the anomalies of the vertical integral of meridional total energy and water vapor flux. As shown in Fig. 3, an anomalously large advection of energy and water vapor from lower latitudes, which is diverted by wind variations, prevailed in the region with conspicuous sea ice retreat (Fig. 1) in spring 2020. Regions around the Laptev and Kara Seas (45° E-120° E, 70° N), i.e., the boundary region between the centers of high (Siberia) and low (central Arctic) pressure, are the main entry channels for warm air-mass input from lower latitudes. It is estimated that the zonal mean of the meridional total energy flux (water vapor flux) through these main entry channels over the entire spring in 2020 reached up to $1.74\times10^{11}$ Wm$^{-1}$ ($1.51\times10^3$ kg m$^{-1}$s$^{-1}$), producing a transport that was 2 (3) standard deviations larger than the 1979-2020 climatology ($-2.71\times10^{11}$ Wm$^{-1}$ ($0.32\times10^3$ kg m$^{-1}$s$^{-1}$)). The pronounced poleward energy and moisture through the entry then converged into the Arctic. As depicted in Fig. 3c and d, the major parts of the ice-retreated shelf seas in spring 2020 are characterized by positive convergence anomalies of the atmospheric moisture and energy transport. Particularly, the magnitude of the total energy and moisture flux convergence anomaly even exceeds 50 Wm$^{-2}$ and $9\times10^{-6}$ kg m$^{-1}$s$^{-2}$, respectively, in the Kara Sea.

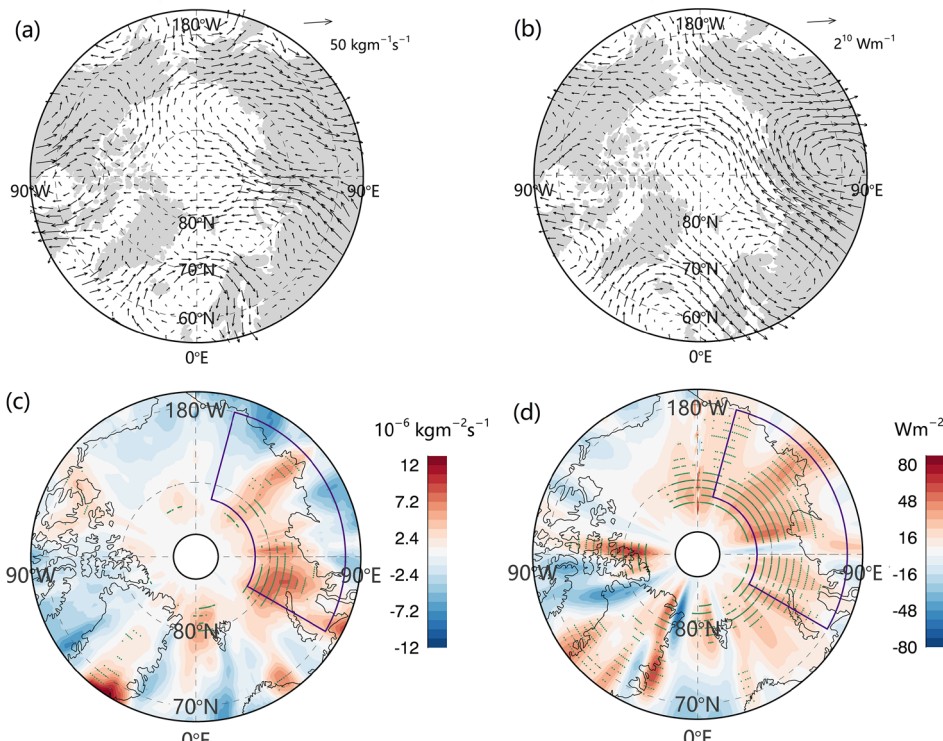

**Figure 3.** Anomalies of the vertically integrated northward (a) moisture flux and (b) total energy transport and the corresponding convergence (c-d) averaged over the spring months (April-June) of 2020. Anomalies are relative to the 1979-2020 climatology. Purple polygons encapsulate areas where substantial sea ice cover loss (60° E-165° E, 70° N-82° N) occurred in July 2020, which represents the study area of this paper. Stipplings represent the values where the anomaly exceeds 2 standard deviations.

Figure 4 illustrates the meridional cross-sections of temperature and specific humidity anomalies spanning the regions with maximum convergence of the atmospheric fluxes (60° E-165° E, 60° N -90° N). Horizontally, elevated temperature and higher moisture content are distributed widely from 60° N to 85° N near the surface. Vertically, the positive temperature and moisture anomalies extend conspicuously into the mid-troposphere (~750 hPa). The intrusion of moisture and energy leads to surface warming (dampening) of up to 3-4 K (6-8×10$^{-4}$ kg kg$^{-1}$) in the spring months. The vertical patterns of the anomalies indicate that the great convergence (Fig. 3c and d) of energy and moisture could contribute to the local increases in the atmospheric temperature and humidity, both at the surface and in the troposphere above the boundary layer, which is in agreement with the finding of Graversen et al. (2008). It is noteworthy that warming and moistening occur over land as well (Fig. 4). The enhanced moisture transport over the land is related to increased precipitation in catchments and increase discharge into the Arctic, which could also favor ice melting (Zhang et al., 2013b) and hinder ice formation in the coastal regions (Dmitrenko et al., 2000). Over Eurasia, springtime temperatures averaging higher than 5 °C resulted in snow melting up to a month early across extensive areas (Ballinger et al., 2020). These processes may also contribute to the minimum SIE of July 2020. We evaluated the vertical profile with land grids being masked and the result shared similar patterns with Figure 4. The consistency confirms that unusual conditions with higher moisture content and warming within the Arctic atmospheric column prevailed over the ice cover loss region. We also examine the role of local evaporation in the regional increase of moisture under a warmer Arctic climate. According to the ERA5 reanalysis, the spring evaporation over the

Arctic Basin exhibits a decreasing trend over the past four decades, except for the Barents and Norwegian Seas. In April-
June 2020, below-normal evaporation dominated the Arctic with an averaged negative value of $-1.5 \times 10^{-4}$ m in the regions
with notable ice-retreat (not shown). The decline in evaporation indicates that the enhanced moisture contributing to the
moister atmosphere over these regions is primarily provided by atmospheric transport from remote areas rather than by local
sources as the moisture fluxes from the sea surface are negligible. This finding further affirms the arguments of Vázquez et
al. (2016) and Singh et al. (2017), which highlighted the importance of remote sources of water vapor for the Arctic.

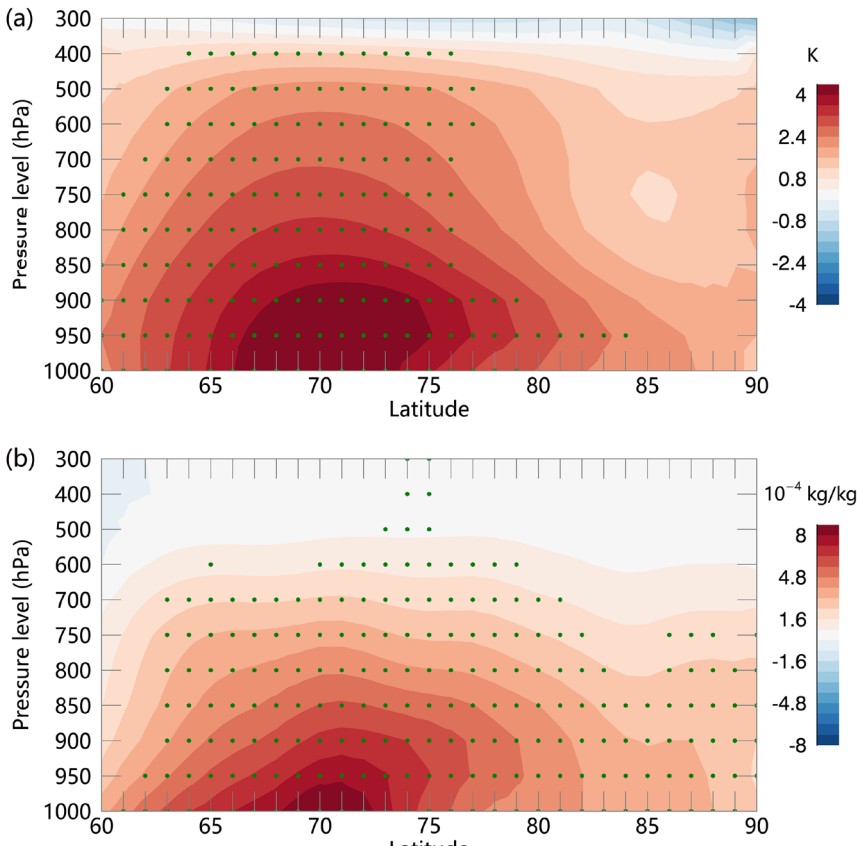

**Figure 4.** Vertical cross-sections of zonal averaged (a) air temperature and (b) specific humidity anomalies, as a function of latitude and
pressure level, during the spring months (April–June) of 2020 spanning the regions with significant energy and moisture convergence (60°
E-165° E, 60° N-90° N). The anomalies are calculated as the difference between the averaged fields of the three months (April-June) and
the corresponding 1979-2020 climatology. Stipplings represent the values where the anomaly exceeds 2 standard deviations.

**4 Surface energy Budget**

The surface energy budget that consists of thermal radiation, solar radiation, and turbulent fluxes is vital for sea ice melt and
growth. An increase of humidity associated with the convergence of moisture flux may strengthen cloud formation
(Johansson et al., 2017), which both contribute to the enhanced local greenhouse effect. In addition, the energy convergence
in the atmosphere may partly be radiated directly to space in the form of longwave radiation, and partly radiated to the sea
surface and turbulently mixed, contributing to the sea ice melt. Having shown the anomalously large convergence of water
vapor and total energy transport in April-June 2020, we will present the variations of different surface energy flux
components in the following. Note that the ECMWF convention for vertical fluxes is positive downwards.

In the Eurasian shelf seas with remarkable sea ice shrinkage, the surface gained more energy owing to both shortwave and longwave radiation, as well as turbulent fluxes, as the enhanced surface fluxes predominantly appeared in these regions (Fig. 5). The spatial pattern of anomalies in surface thermal radiation downwards is characterized by positive values throughout the convergence zones, with the largest amplitudes in the Kara Sea (~32 Wm$^{-2}$, Fig. 5a). The anomalies of the net longwave radiation (Fig. 5b) are roughly similar in spatial distribution to that of the downward component over the Arctic marginal seas on the Eurasian side. The difference between Fig.5a and b indicates that part of the thermal radiation was radiated upwards to increase the surface air temperature before the melt commenced. The downward component of the solar radiation was below-normal in most parts of the ice-retreated area (Fig. 5d), which is presumably attributed to increased cloudiness associated with the convergence of moisture (Johansson et al., 2017). In contrast, positive anomalies of the net solar radiation were found in the Eurasian shelf seas where the extensive loss of sea ice was observed. This is a result of the substantial fraction of open water due to sea ice loss which reduces the albedo and thereby enables the upper ocean to absorb more heat (i.e., the ice-albedo feedback).

Additionally, sensible and latent turbulent surface flux anomalies both contributed to the energy surplus at the surface in the spring months of 2020 (Fig. 5c and f). The positive (downward) anomalies of turbulent surface fluxes were detected in the regions with contracted ice cover (Fig. 5c and f). Intuitively, more turbulent fluxes would be released to the atmosphere as more open water prevailed. That is, a negative (upward) value over the Arctic shelf seas is expected. However, reduced upward, or even downward, sensible and latent heat fluxes were detected in the study region during April-June in 2020. This can be attributed to the anomalously high moisture advection and convergence which could reduce the gradient of the water vapor pressure at the surface. As implied in Fig.4, positive temperature and humidity anomalies extended from surface even to mid-troposphere, peaking at around 925 hPa. These changes would result in a decreased vertical gradient in air temperature and humidity in the lower atmosphere, reducing the hypothesized upward turbulent fluxes from the ocean surface to the overlying atmosphere. Note that even if there exist discussions about the radiative effects of cloud (Doyle et al., 2011; Serreze and Barry, 2011; Babar et al., 2019), measurements for clouds are largely model generated, leading to large uncertainties in reanalysis and in quantifying radiative fluxes. The cloud radiative effect biases can be attributed to errors in cloud properties, including inadequate Arctic cloud amount (English et al., 2015); inaccurate partitioning of cloud water phase (Cesana et al., 2012; Kay et al., 2016); and insufficient supercooled liquid clouds (Komurcu et al., 2014). However, it is understood that clouds warm the surface except for a brief period in summer as the surface net all-wave radiation is larger in the presence of clouds (Serreze and Barry, 2011).

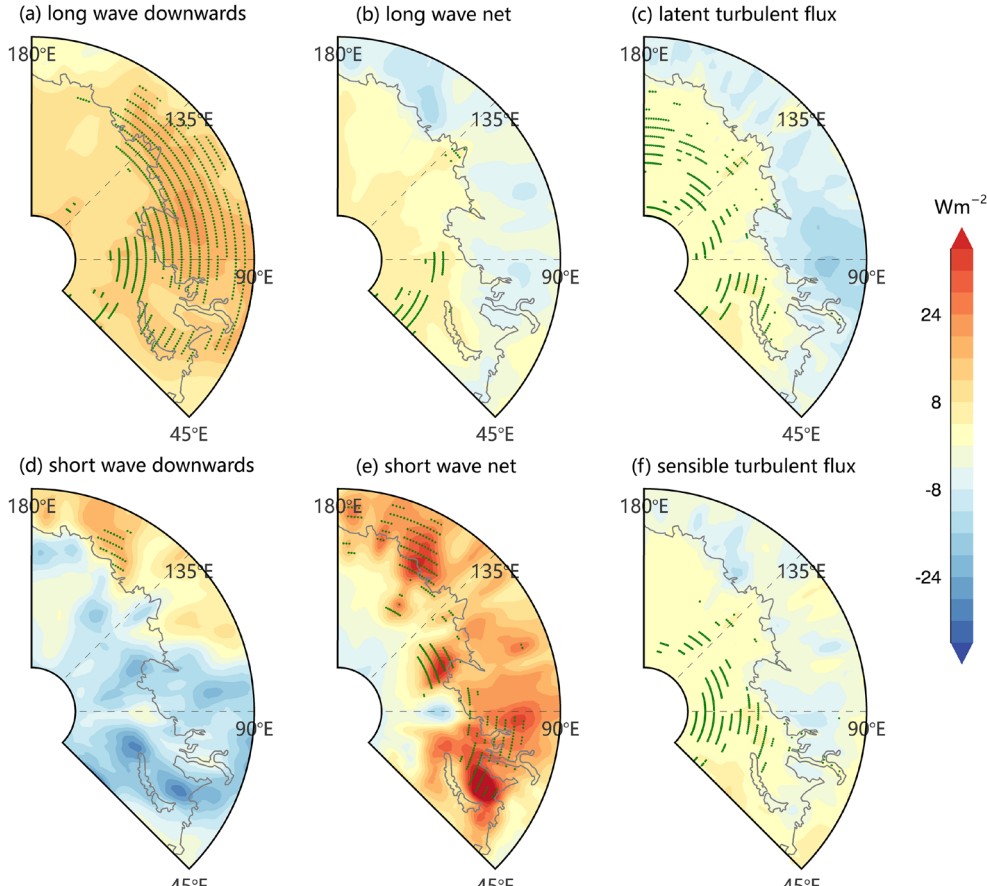

**Figure 5.** Anomalies of surface (a) downwelling and (b) net longwave radiation, (d) downwelling and (e) net shortwave radiation, as well as sensible (c) and latent (f) heat fluxes. The anomalies are calculated as the difference between the averaged fields of the three months (April-June) and the corresponding 1979-2020 climatology. The stippled grids denote those with anomalies exceeding 2 standard deviations. Note that the plotted view is slightly larger than the study area for better illustration.

Figure 6a presents the time series of SIE, the anomalies of atmospheric energy transport convergence and the surface fluxes averaged over the study area (enclosed by the purple polygon in Fig. 3c and d) during 2020. Indeed, an energy convergence event started at the end of March and lasted for three months, peaked in early June (Fig. 6a, grey line). The convergence event is intense with values higher than the climatology in April and May by two standard deviations. Not that noticeable convergence of moisture was also observed in spring 2020 albeit not as prominent as the energy convergence event (not shown). This is followed by an augment in the downward thermal radiation plus turbulent fluxes (smaller) by ~30-40 Wm$^{-2}$ (Fig. 6a, green line). There were 51% of the days in spring 2020 having unusually positive downward thermal radiation plus turbulent fluxes (one standard deviation above the 1979-2020 climatology) in the study area. The almost simultaneous change of downward thermal radiation highlights that the convergence of the total energy and moisture flux has a significant imprint on the increased surface energy fluxes. With the enhanced downward infrared radiation, sea ice cover began to decrease gradually (Fig. 6a, blue line). The temporal development in Figure 6 underlines that the positive anomalies of longwave radiation plus turbulent fluxes played a significant role in initiating an earlier melt onset in 2020. Besides, the moisture transport from lower latitudes into the Arctic could be related to an increase in rain, which enhances

surface melt leading to ponding (Vihma et al., 2016). As estimated from ice surface melt dates archived in NASA, persistent

melt conditions in the study area were observed in May 2020 (Fig. 6a, pink vertical line), which occurred about 15 days

earlier than the 1979-2020 average (Fig. 6b). As the surface melt commenced, the decreased surface albedo in turn acted to

increase the absorption of solar radiation by the ocean-ice system (Fig. 6a, red line). That is, the earlier ice surface melt onset

could foster stronger ice-albedo feedback (Hall, 2004), leading to an accelerated decline in SIE in June-July when the

300 anomaly of net solar radiation reached its maximum. Likewise, several previous studies point out that earlier melt onset of

Arctic sea ice is associated with the intensified northward transport of moist and warm air. Enhanced moisture and energy

flux convergence result in positive anomalies of air temperature, precipitable water vapor, and cloud fraction, which can

increase the downwelling longwave radiation to the surface (Mortin et al., 2016; Liu and Schweiger, 2017; Huang et al.,

2019; Horvath et al., 2021). Our results augment evidence for the existing knowledge of driving mechanisms of early melt

onset, especially for the Arctic peripheral sea, where the rain-on-ice events have been more and more frequently observed

(Dou et al., 2021), and the melt pond coverage is increasing (Lei et al., 2016).

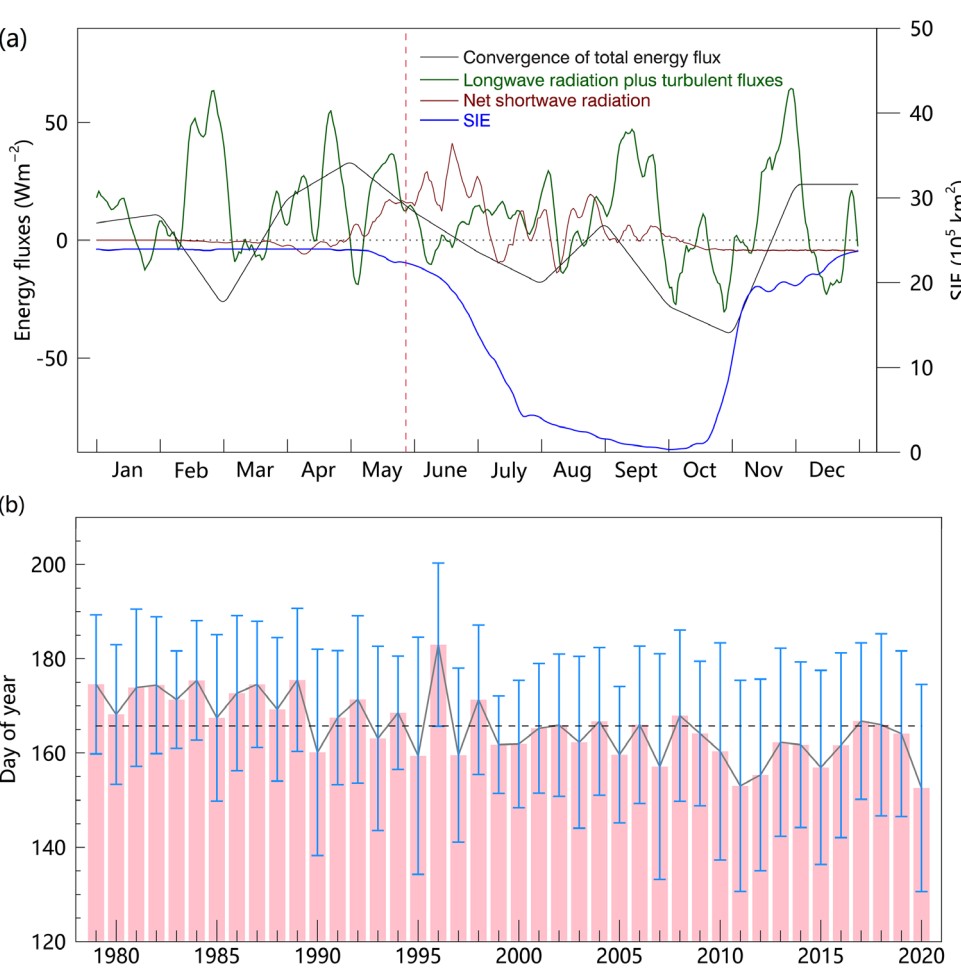

**Figure 6.** (a) Time series of SIE, the anomalies of atmospheric energy transport convergence and surface energy fluxes over the study area
(indicated by the purple polygon in Fig. 1) during 2020. The blue curve represents the SIE. The red line denotes the anomalies of net solar
radiation. The green line corresponds to the anomalies of the sum of the downwelling thermal radiation and the turbulent (latent plus sensible)
flux. The vertical pink dashed line denotes the average date of ice surface melt (May 28) in 2020. The retrieved values are averaged over the
oceanic grids by applying a land mask. The anomalies are relative to the 1979-2020 climatology of the years. (b) Melt onset date in each

year averaged over the study area during the period 1979-2020. The grey dashed line represents the mean melt date of these four decades. Error bars denote one standard deviation.

We calculate the changes in sea-ice thickness due to the variations of surface energy fluxes via a simplified sea-ice growth model (Parkinson and Washington, 1979), so that we can quantify the thermodynamic impact of atmospheric energy of spring 2020 on the sea ice melt. According to equation (4), a 1 W/m$^2$ increase in surface energy budget during three spring months (April to June) would melt approximately 2.60 cm of sea ice. The spatial pattern of sea ice thickness change due to surface energy fluxes variations, calculated by equation (4), is portrayed in Fig. 7a. SIT anomalies due to radiative forcing are mostly

negative (i.e., melting) in the Kara, Laptev, and East Siberian Seas during the three spring months of 2020, with a particularly large value (-1.2m) in the Kara sea (Fig.7a). The region with significant SIT reduction (wherein sea ice was totally melted) agrees well with that with distinct SIC anomalies. Moreover, to reiterate the long-term changes, we examine the trend of SIT over the past four decades in the study area. As estimated from the thickness data provided by PIOMAS, the average thickness of spring (April-June) sea ice in the study area has a remarkable decreasing trend of -0.27 m per decade (significant at the 99%

confidence level, Student's t-test) in the past four decades (Fig. 7b). SIT was persistently lower than 2.50 m since 2000 and dropped sharply to only 1.20 m in the spring of 2020. Thinner ice is more susceptible to changes in thermodynamic forcing, thus prone to melt earlier, which in turn could foster a stronger summer ice-albedo feedback through the presence of open water areas. In other words, without the extensive coverage of thin, first-year ice in spring 2020 in the study area (Fig. 7b), the unusual atmospheric energy and moisture transport would not have been nearly as effective in reducing ice extent as was

observed (Fig. 1).

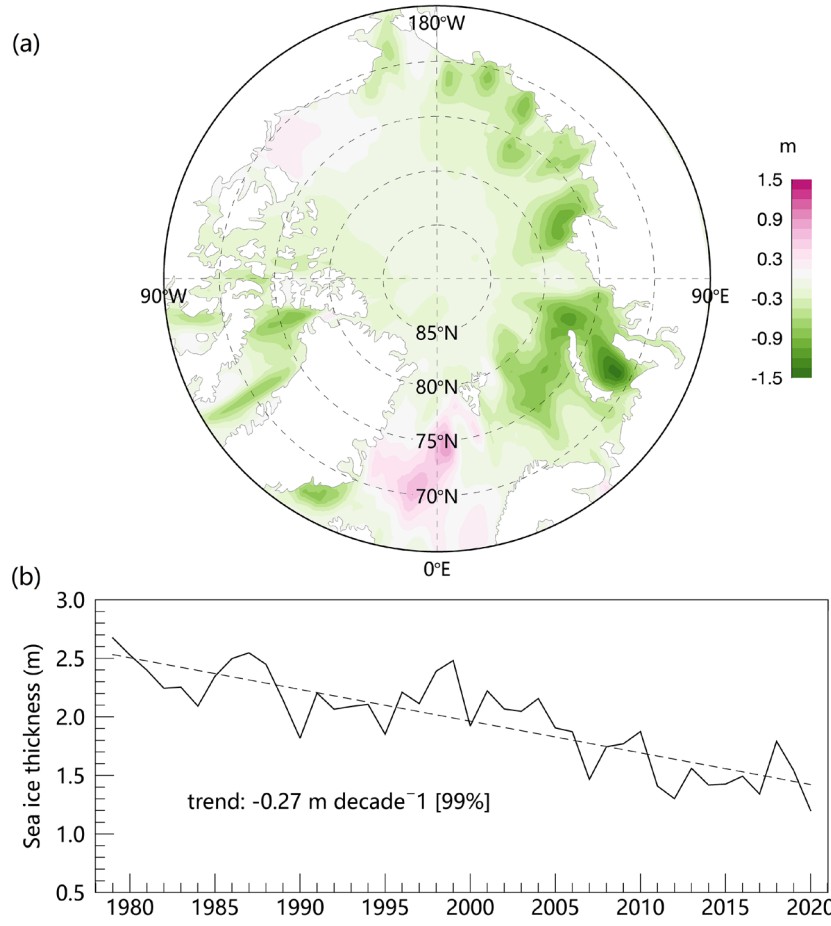

**Figure 7.** (a) Changes of SIT caused by anomalies of surface radiative fluxes during spring (April-June) 2020 which is estimated by a simplified sea-ice growth model. (b) Time series of SIT (provided by PIOMAS) and the corresponding trend (dashed line) averaged over the study area in spring during the period 1979-2020.

## 5 Cyclones activities in Spring 2020

Synoptic cyclones are a central component maintaining the global atmospheric energy, moisture, and momentum budgets (Jakobson and Vihma, 2010; Dufour et al., 2016; Villamil-Otero et al., 2018). A wide variety of studies reveal a poleward shift in tracks and significant changes in the frequency and intensity of extratropical cyclones (Yin, 2005; Sepp and Jaagus, 2011). Considering the notable variations in cyclone activities and the strong sea ice decline experienced in the study area over recent decades, understanding the underlying effects of the cyclones on the poleward transport of energy/moisture is especially crucial. We identify and track cyclone systems that occurred in the spring months (April - June) during the period 1979-2020 using the automated algorithm (Serreze et al., 1993; Wang et al., 2013).

Longitudinal distributions of the climatological vertically integrated northward total energy and moisture flux across 60° N, as well as the CAI of cyclones entering the Arctic at the 60° N averaged over the spring months (April–June) during 1979-2020, are illustrated in Fig. 8. Note that cyclones entering the Arctic are defined as cyclone trajectories having their cyclolysis south of 60° N and traveling poleward. The spatial distribution of CAI is in good agreement with the vertically

integrated meridional total energy and moisture flux. The main entry channels of the energy and moisture including the North Atlantic, North Pacific, and the Labrador Seas witness more cyclones with greater intensity that propagated toward the Arctic region. A strong correlation exists between the averaged CAI and the vertically integrated northward total energy flux (moisture flux) at 60° N with R=+0.69 (+0.68) (361 grids, significant at the 99% confidence level), suggesting the significant role of cyclone activity in contributing to the poleward advection of energy and moisture. Note that Greenland is masked when tracking cyclones to avoid problems caused by SLP extrapolation, hence we use the latitude 60° N rather than 70° N to display the relationship between cyclone activities and the meridional fluxes. Other studies also corroborated the fact that synoptic cyclones play a crucial role in regulating the poleward fluxes considering the fundamental nature of cyclones in holding and transporting moisture and energy. For instance, Dufour et al. (2016) concluded that poleward advection of moisture is dominated by transient eddies (e.g., cyclones) owing to its almost exclusively meridional direction of the flux. It was estimated that the cyclones could explain 80–90% of the total northward transport at latitudes of 70° N (Jakobson and Vihma, 2010; Dufour et al., 2016). In addition, Villamil-Otero et al. (2018) found that stronger cyclone activity across 60° N measured by the CAI generally co-occurs with enhanced poleward monthly atmospheric moisture transport in all months. Our results augment evidence for the view that the intrusion of moisture and energy associated with cyclones into the Arctic can be linked to the abrupt changes in the Arctic climate system.

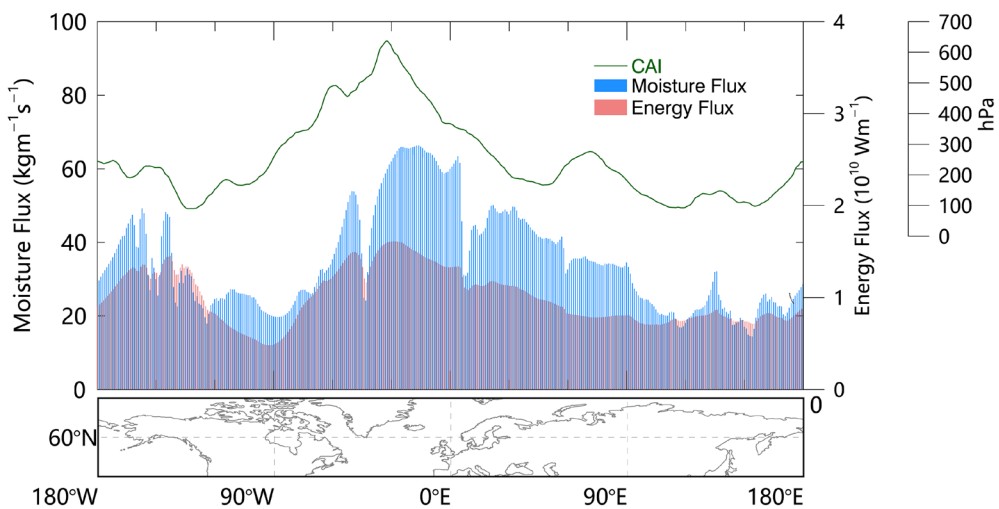

**Figure 8.** Longitudinal distributions of the climatological vertically integrated meridional total energy and moisture flux across 60° N, as well as the CAI of cyclones entering the Arctic at the 60° N averaged over the spring months (April–June) during 1979-2020.

It is noteworthy that, in this study, we use a range of latitudes (50° N-70° N, with a step length of 1°), other than a single one, to define the poleward cyclones. For instance, poleward cyclones are defined as those that are generated south of a certain latitude within the range (50° N-70° N) and traveling northward through it. All of these cyclones may play a non-negligible role in carrying energy and water vapor to the Arctic in the form of a relay. As shown in Fig. 9, spring 2020 saw many low-pressure systems moving poleward from Eurasia and some of them entered the study area through the main entry channels in the Kara Sea (Fig. 9, green thin lines). Besides, in the Eurasian shelf seas with great convergence of the total

energy and water vapor transport (Fig. 3c and d), the majority of the cyclones are featured with trajectories in a zonal direction (Fig. 9, blue thin lines). Furthermore, we retrieve the typical trajectory paths of these cyclones following Gaffney (2004). The trajectory clustering was done using a polynomial regression mixture model where each cyclone trajectory is

approximated as a second-order polynomial. The detected cyclones in spring 2020 are clustered in two categories, which are schematically explained in Fig. 9 with thick polylines. One track represents the cyclones that are generated in the lower latitude of Eurasia with a poleward moving tendency, while the other denotes cyclones in the marginal seas that are characterized by an eastward movement toward or through the Kara, Laptev, and East Siberian Seas (Fig. 9). In general, the trajectories of these cyclones as observed during spring 2020 coincide well with the path of total energy and water vapor

transport (Fig. 3a and b). The good agreement implies that these extratropical cyclones in spring, as shown in Fig. 9, served as a vital carrier of the anomalously large amount of energy and moisture into the study area. To sum up, the synoptic cyclones act in concert with the large-scale atmospheric circulation to cause anomalous energy and moisture fluxes into the study area and to change the characteristics of the Arctic climate system in 2020.

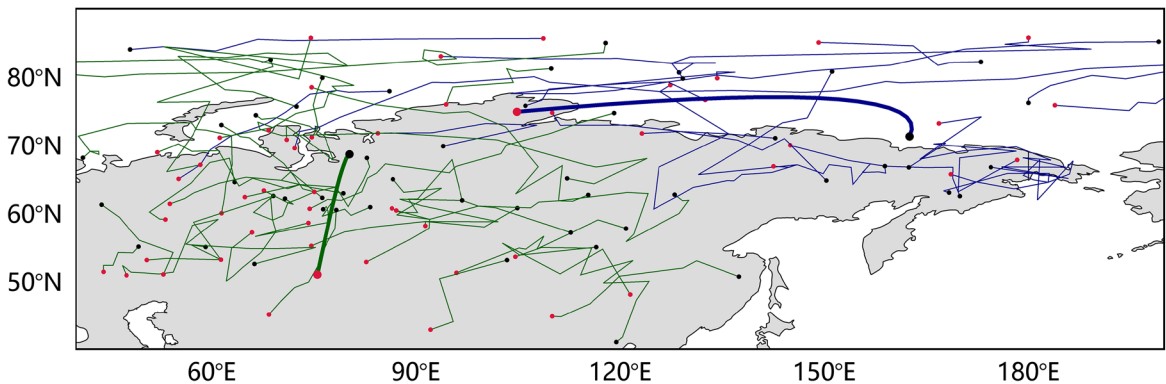

**Figure 9.** Poleward cyclones detected in the main entry channels (green thin line, 45° E-120° E and 40° N-90° N) and all cyclones occurred in the regions intersected with the route after the fluxes entering Arctic (blue thin line, 45° E-15° W and 65° N-90° N) during spring (April-June) in 2020 and their typical trajectories (thick green and blue lines). Red (black) dots represent the position of the genesis (lysis) of cyclones.

We further investigate the connection between the long-term changes in poleward cyclones and meridional transport of

total energy and moisture. Fig. 10a illustrates the decadal relationship between the 10-year running trends that have been observed in the meridional total energy/moisture transport and the poleward cyclone activities in spring during 1979-2020. Note that the northward transport is the average value of all the corresponding fluxes across 50° N-70° N, which is consistent with the definition of poleward cyclones. Indeed, a robust correlation exists between the trends of the average intensity of poleward cyclones and the vertical integral of northward energy (moisture) transport during spring with a strong correlation

coefficient of +0.62 (+0.59), suggesting poleward cyclone activities play an important role in regulating the variations of the decadal trends in meridional transport of energy and moisture. Particularly, in the recent decade (2010-2020), significant upward trends are observed in the northward transport of total energy and moisture together with more intense poleward

cyclones (Fig. 10a). It is noteworthy that the 10-yr trends of these three parameters in spring exhibit remarkable low-frequency variation, which is presumably controlled by the large-scale circulation trends. The low-frequency oscillation of moisture transport trends is primarily linked to the trends in the northward component of surface wind as wind and water vapor gradient determined the flux. We found that when poleward cyclone intensity and meridional total energy/moisture transport exhibit decreasing trends, the large-scale atmospheric circulation shows a tendency towards a stronger anticyclonic circulation over Greenland and Eastern Siberia with a barotropic structure in the troposphere (not shown), similar to the pattern found in Ding et al. (2017). When the opposite holds, greater geopotential height extending from surface to upper-level tends to emerge in the Norwegian Sea and Chukchi Sea (figures omitted).

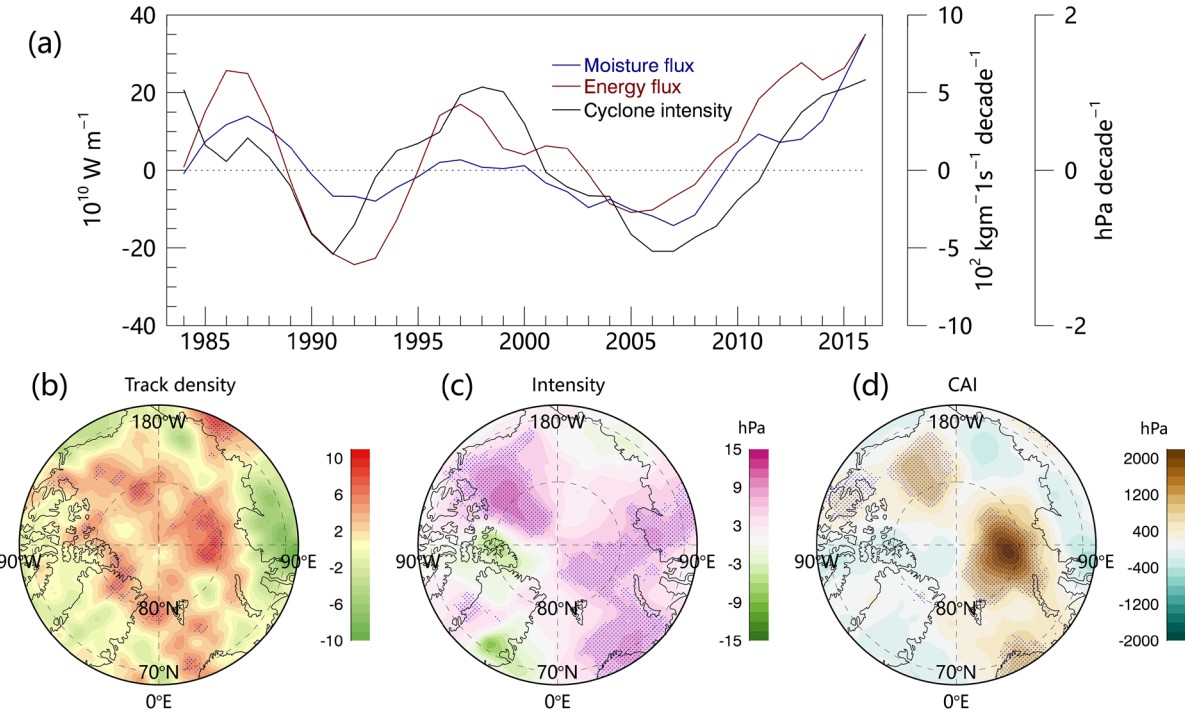

**Figure 10.** (a) The time series for the trends of the meridional total energy (moisture) transport and average intensity of poleward cyclone during spring months (April to June) from 1979 to 2020. The trends are calculated using a 10-year running window. Note that the northward transport is the average value of all the corresponding fluxes across 50° N-70° N, which is consistent with the definition of poleward cyclones. The values of the x-axis correspond to the central years of each running window. Anomalies of the cyclone features in spring 2020 relative to the 1979-2020 climatology, including the (b) density of tracks, (c) intensity, and (d) CAI. Panels have units of counts per $10^6 \, \text{km}^2$ (b) and hPa per $10^6 \, \text{km}^2$ (c and d). Blue dots represent the regions with values above the mean plus 1.5 standard deviations.

As for spring 2020, stronger and more frequent cyclones were detected in the Arctic (Fig. 10b and c). The density of cyclone tracks was higher than normal in many parts of marginal seas and the central Arctic Ocean, with the largest positive values centered over the Taymir Peninsula (Fig. 10b). Most cyclones throughout the Arctic Basin had unusually high intensity than the climatology of the years 1979-2020, especially in the Barents, Kara, Laptev, and Beaufort Seas, indicating lower-than-normal SLP in these regions (Fig. 10c). The spatial pattern of CAI anomalies was roughly in line with those of track density and intensity (Fig. 10d). In general, the Eurasian shelf seas had more frequent and stronger cyclones in spring 2020, especially in the Kara and Laptev Seas (Fig. 10). The cyclone variations could alter the spatiotemporal characteristics

of the critical near-surface atmospheric parameters (wind stress, temperature, and humidity). As a result, the atmospheric anomalies could exert a significant impact on sea ice in the study area through the regulation on ice motion, deformation, and melt associated with both dynamic and thermodynamic processes.

From the thermodynamical view, the enhancement of the total energy and water vapor transport in the Eurasia side (Fig. 3a and b) is associated with the regional increases both in the number (Fig. 10b) and intensity (Fig. 10c) of synoptic cyclones that occurred in the main entry channels and some parts of the study area. As analyzed above, the warm and moist air masses carried by cyclones in spring (Fig. 9) could alter the surface energy balance, thus initiating the earlier melt onset of sea ice as observed in the study area. Persson (2012) also found that the synoptic-scale weather systems that augmented the atmospheric energy fluxes to the surface can trigger the melt onset at a specific site and a certain year. The cyclones and the associated frontal systems can also affect the formation of low-level and midlevel clouds over the Arctic Ocean (Curry et al., 1996), which can alter the surface radiation and energy budgets. Moreover, the cyclones traversing the Arctic can trigger a spatially extensive sea ice melting with their associated frontal systems (Stramler et al., 2011). Zhang et al. (2013a) found that an intense cyclone in August 2012 led to an accelerated ice volume decrease, owing largely to a quadrupling in bottom melt caused by increased upward ocean heat transport. Thinner initial ice thickness preconditions make sea ice more sensitive to synoptic events. All these thermodynamic factors may contribute to the significant SIT decline of spring 2020 in Eurasia shelf seas as shown in Fig. 7.

Thorndike and Colony (1982) revealed that surface wind can explain more than 70% of the variance of the ice velocity in the central Arctic Ocean. The extreme loss of SIE in July 2020 was accompanied by a strong pattern of anomalous cyclonic SIM in the previous spring (Fig.11), with Ekman drifting out of the central Arctic toward the marginal seas and ice outflow through the Fram Strait. A positive trend of $+3.70\times10^4$ km$^2$/decade (significant at the 99% confidence level, Student's t-test) is observed in the spring sea ice area flux through the Fram Strait during 1979-2020. In spring 2020, the sea ice areal flux was $2.75\times10^5$ km$^2$ as estimated following Kwok (2007), which barely departed from the linear trend. The unexceptional ice outflow through Fram Strait during spring 2020 excludes dynamical transport of ice to be the primary contributor to the occurrence of July minimum SIE. On one hand, the cyclonic SIM anomaly in cold seasons serves to enhance the production of new ice within leads because of the increase in sea ice divergence (Krumpen et al., 2021). On the other hand, as the melt season commences, while ice divergence increases extent, it can also accelerate melt by exposing more dark open water areas in the cracks, leads, and polynyas. Due to the dynamics of cyclones, strong and inhomogeneous storm-induced wind anomalies can change the sea ice drift pattern and deform the ice cover. More frequent and intense cyclones in the Arctic during spring 2020 (Fig. 10) may provide additional cyclonic wind anomalies which are superimposed on that of the large-scale atmospheric circulation as depicted in Fig. 2, promoting the above processes. Based on our results, the thermodynamical other than dynamical effects of cyclones seem to play a dominant role in regulating the SIE changes in

the study area during spring 2020, as the expansion of sea ice cover due to divergence was offset by the significant shrinkage due to melt.

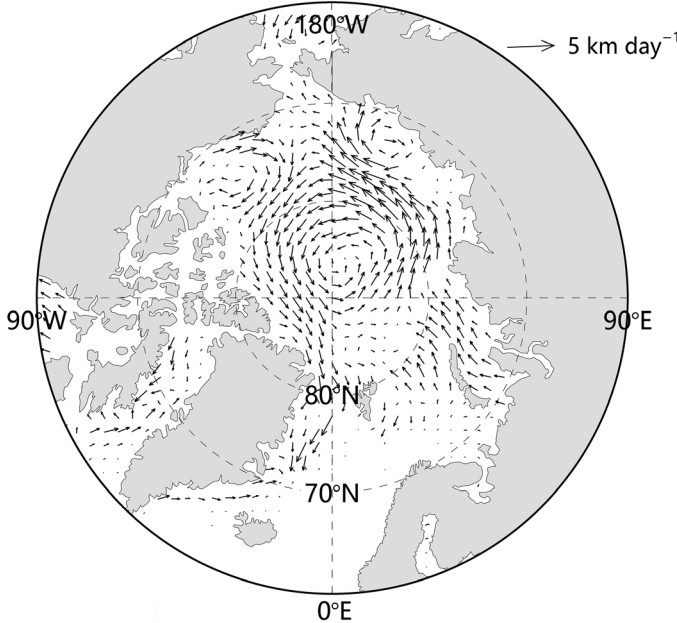

**Figure 11.** Anomalies of the spring (April to June) SIM in 2020 relative to the climatology of the years 1979-2020.

## 6    Discussion and Conclusions

An unprecedented reduction in SIE was observed in July 2020 since the satellite era (1979-2020), especially in the Eurasia shelf seas including the Kara, Laptev, and East Siberian Seas (60° E-165° E, 70° N-82° N). By utilizing global reanalysis datasets and satellite observations, we address the underlying mechanisms of the extreme event. The variations of the total energy and moisture transport toward the study area are obtained and analyzed. We investigate the associated surface energy budget during spring (April to June) of 2020 to disentangle the driving effects of different energy components on sea ice in July. Moreover, the influences of large-scale atmospheric circulation and synoptic cyclones on the poleward energy and moisture transport are outlined.

Our results reveal that anomalously high advection of energy and water vapor prevailed during spring 2020 over the regions where conspicuous sea ice retreat occurred in the following July. The enhanced energy and moisture flux converged into the study area through the main entry channels in the Laptev and Kara Seas from lower latitudes, which reached up to $1.74 \times 10^{11}$ Wm$^{-1}$ and $1.51 \times 10^{3}$ kg m$^{-1}$s$^{-1}$, respectively, over the entire spring. As a consequence, the convergence of the transport increased the temperature and specific humidity of the local atmosphere from the surface to the middle-troposphere. This is accompanied by a strengthened downward longwave radiation plus turbulent fluxes at the surface, which initiated the earlier melt onset of sea ice in the study area (15 days in advance compared to the climatology). After the melt commenced, the enhanced net solar radiation absorbed by the ocean-ice system produced an accelerated decline in SIE through the ice-albedo feedback. Quantitative analysis shows that the amount of surface radiative fluxes surplus (~40 Wm$^{-2}$)

during April–June 2020 in the ice-retreat domain can potentially melt around 1 meter of ice in addition to the climatological melt. Besides, having experienced a large reduction in thickness during recent decades (-0.27 m per decade), the majority of the present sea ice in the study area is composed of thinner seasonal ice (Kwok, 2018). We conclude that the fact of younger and thinner sea ice, together with enhanced total energy and moisture transport which affect the surface radiative forcing, has repercussions for the occurrence of the record low SIE in July 2020.

A key driver of the anomalous high transport of the total energy and moisture during spring 2020 was a persistent atmospheric pattern, featuring unusually low SLP over the north pole which extended through the Barents-Kara Sea to Eurasia and unusually high-pressure centers over the Eastern Siberia and the Norwegian Sea. The SLP pattern led to southerly winds and favored the enhanced transport of warm and moist air mass from Eurasia to the adjacent Arctic shelf seas where substantial sea ice retreat was observed in July 2020. Besides, the typical trajectories of the synoptic cyclones that occurred on the Eurasian side in spring 2020 agree well with the path of the intensive total energy and water vapor transport. The agreement implies that cyclones served as another important carrier of the large energy and moist fluxes into the study area since the storm is capable of holding considerable moisture and energy. Further analysis reveals that the enhanced atmospheric fluxes in spring may be partly attributed to the stronger and more frequent cyclone activities near the region with severe loss of SIE. Moreover, anomalously frequent and intense cyclones in the Arctic during spring 2020 coupled with large-scale atmospheric circulation, further strengthened the cyclonic wind and ice motion. The cyclonic ice drift could lead to extensive sea ice melt in the study area as presented in July 2020 through the large formation of the cracks and leads among sea ice floes. Here we argue that the unusual atmospheric energy and moisture transport favored by large-scale circulation coupled with cyclones in Spring 2020 effectively reduced ice extent under the circumstance of more thin ice in the study area. In the present study, we explored the influence of cyclones qualitatively because of the strong coupling between the large-scale atmospheric circulation and synoptic activities (Cohen et al., 2017; Koyama et al., 2017). The coupled interaction between sea ice and atmosphere involves myriad physical processes which may lead to diverse and nonlinear effects on the Arctic ice, as well as triggering multiple feedback mechanisms. Disentangling their effects thereby is challenging which requires using more sophisticated statistical techniques, combined with climate models of higher fidelity. Under the circumstance of thinner and younger sea ice in the Arctic, a comprehensive and quantitative analysis of different mechanisms associated with cyclone activity would be an appropriate avenue for follow-up research.

It is also noteworthy that 2020 had the second-lowest September SIE since the satellite era, which is merely 12% higher than that of 2012. A prominent high-pressure anomaly dominated the Arctic in July-September 2020 (especially in July-August), with a high-pressure center slightly shifted to the Pacific sector of the Arctic. Previous studies have elaborated that the recent summertime sea ice depletion is broadly associated with the anticyclonic atmospheric circulation pattern which can increase the downwelling longwave radiation above the ice by warming and moistening the lower troposphere (Ogi and Wallace, 2012; Ding et al., 2017). Common to existing literature, the temperature and specific humidity of the region under

the control of the high-pressure system in July-September 2020 were notably great than the corresponding climatology. That is to say, the combination of low-pressure anomaly persistent in April-June (favoring moisture and energy inflow) and anticyclonic atmospheric circulation pattern (leading to adiabatic warming) might contribute to the particularly low SIE of September 2020. Besides, positive SIC anomalies were detected in the eastern Beaufort Sea in summer 2020. The increasing SIC herein can be attributed to the anticyclonic winds, since it could drive older sea ice, which has more opportunity to survive through the summer melt, from the central Arctic Ocean and the Canadian Arctic Archipelago into the Beaufort Sea. The multiyear ice transport might partially retard the rapid melt of the overall sea ice in the Arctic after late July and hindered 2020 from being a new record.

**Data availability**

NSIDC sea ice motion is available at https://nsidc.org/data/NSIDC-0116/versions/4 (last access on 7 July 2021). NSIDC sea ice concentration data is obtained from https://nsidc.org/data/NSIDC-0079/versions/3 (last access on 7 July 2021). Sea ice thickness from PIOMAS is downloaded from http://psc.apl.uw.edu/research/projects/arctic-sea-ice-volume-anomaly/data/model_grid (last access on 23 March 2021). Sea ice surface melt dates are available at https://earth.gsfc.nasa.gov/index.php/cryo/data/arctic-sea-ice-melt (last access on 22 February 2021). The ERA5 reanalysis datasets were obtained from the website https://cds.climate.copernicus.eu/cdsapp#!/search?type=dataset (last access 20 December 2021).

**Author contribution**

YL carried out the analysis, processed the data, and wrote the manuscript. HB designed the method to calculate the moisture flux and performed experiments. YW processed the sea ice datasets. HH contributed to the ERA5 data collection. All authors contributed to the discussion and provided ideas during the concept phase and the writing process.

**Competing interests**

The authors declare that they have no conflict of interest.

**Acknowledgments**

This work was supported by the General Project of Natural Science Foundation of Shandong Province (ZR2020MD100), National Natural Science Foundation of China under Grant (41406215, 42076185, and 41976219), Key Deployment Project of Centre for Ocean Mega-Research of Science, Chinese Academy of Sciences (COMS2020Q12), the Open Funds for the Key Laboratory of Marine Geology and Environment, Institute of Oceanology, Chinese Academy of Sciences (MGE2020KG04)

and Academy of Finland (Grant No. 317999).

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
