# Peer review of "Contribution of warm and moist atmospheric flow to a record minimum July sea ice extent of the Arctic in 2020."

_The Cryosphere, 2021_

## Author Comment (AC2)

**Response to Referee #2**

Review of the Manuscript 'Warm and moist atmospheric flow caused a record minimum July sea ice extent of the Arctic in 2020' by Ling et al. submitted to The Cryosphere.

**Summary:**

Liang et al. aims to investigate the July 2020 extreme sea ice melt event in terms of physical mechanisms. They look at the prior late spring-early summer 2020 to explain that anomalous warm air intrusion and cyclone activity set up favorable conditions for sea ice melt in July 2020. I find the idea interesting and well suited for The Cryosphere journal and the methods generally appear sound, however the presentation of their results and the significance of the findings need a bit more elaboration before I could recommend the paper for publication.

**Reviewer comments**

**R.1.** I find the Introduction a bit hard to follow. The authors might consider reorganizing it a little bit via discussing the contents of the current second paragraph before starting to talk about the 2020 SIE extent and referring to Figure 1. From row 30 it reads like it is already the description of the Results. I understand the reasoning behind it; the authors want a succinct Introduction to go with their very specific and well-defined goal in the paper, however I think they could do better in setting up the research question.

Especially, I suggest that the authors discuss more thoroughly the current understanding of oceanic and atmospheric drivers of summer sea ice melt, especially the physical mechanisms, as their objective in this paper is to reveal the underlying mechanisms leading to the record melt in July 2020. For example, in the current introduction the authors only mention surface wind driven sea ice drift as dynamical forcing on sea ice, however in recent years anticyclonic circulation anomalies caused vertical motion (warming and moistening descending air) is also a key component of atmospheric forcing on sea ice (see e.g., Ding et al. 2019; Topal et al. 2020). This local atmosphere-sea-ice coupling mechanism is further linked to large-scale circulation changes and forcing from the tropics especially over the enhanced melt period between 2007 and 2012 (Screen and Deser 2020; Warner et al. 2020; Baxter et al. 2019). Therefore, the well-known thermodynamical factors causing sea ice melt may be better linked with known dynamical sources

besides surface wind drift, which is far from being the only dynamics causing sea ice variations in the Arctic. In this way the authors may set up their research question a bit more connected to existing literature and highlight that their goal is to complement the existing knowledge of dynamical drivers of sea ice loss which can well be exemplified via a case study in July 2020.

June-August 2020 was dominated by a high-pressure anomaly in the Arctic, which could have acted in concert with the prevailing spring conditions to cause the sea ice extreme melt. I wonder if the authors could provide more discussion on how they distinguish their results or link together with previous literature either in the Introduction or in their Discussion part.

Response: We sincerely appreciate the constructive and detailed comments by Referee #2. These comments helped us improve our manuscript, and provided important guidance for our future research.

a. The Introduction will be reorganized as suggested. Before analyzing the severe event of sea ice loss in July 2020, we will discuss thoroughly the current understanding of atmospheric drivers of sea ice melt, especially the relevant physical mechanisms and refer to the previous studies. The scientific question of the present research will be set up afterward.

b. Although the September SIE of 2020 did not shatter the previous lows to be a new record, September 2020 had the second-lowest SIE since 1979, stood at $3.74 \times 10^6$ km$^2$, which is merely 1% higher than the lowest SIE. A prominent high-pressure anomaly dominated the Arctic in June-August 2020 (especially in July-August, the figure below). Previous studies elaborated that the recent summertime sea ice depletion is broadly associated with the anticyclonic atmospheric circulation pattern which can increase the downwelling longwave radiation above the ice by warming and moistening the lower troposphere (Ogi and Wallace, 2012; Ding et al., 2017; Bi et al., 2021). This kind of variation in local atmospheric circulation patterns is further linked to large-scale circulation changes and forcing from the tropics through teleconnections (Baxter et al., 2019; Screen and Deser, 2019; Warner et al., 2020). The combination of low-pressure anomaly persistent in April-June (favoring moisture and energy inflow) and anticyclonic atmospheric circulation pattern (leading to adiabatic warming) may contribute to the particularly low SIE of September 2020, the mechanisms of which would be the potential candidates for future studies. The present study is dedicated to elucidating that anomalous high inflow of total energy and moisture from lower latitudes to

the Arctic in spring caused severe sea ice loss of July 2020. The above arguments about the anticyclonic atmospheric circulation pattern therefore will be added to the Discussion and Conclusions part when mentioning the September SIE of 2020.

[Figure]

**Supplementary Figure.** Spatial patterns of sea level pressure anomalies (shading) during July to August 2020. The anomalies are computed as the difference between the averaged fields of the three months (April-June) and the corresponding climatology over the past four decades (1979-2020). Stipplings represent the values where the anomaly exceeds 1.5 standard deviations.

c. Several existing literature pointed out similar mechanisms with ours (Graversen et al., 2011; Vázquez et al., 2017; Kapsch et al., 2019; Horvath et al., 2021). Our results serve to augment more evidence to the mechanisms that drive sea ice loss through transporting moisture and energy into the Arctic via a case study in July 2020. Here we argued that the unusual atmospheric energy and moisture transport favored by large-scale circulation and cyclones in Spring 2020 effectively reduce ice extent under the circumstance of more thin, first-year ice, which is a novel result. These points distinguishing our research from previous literature will be added to the revised version.

**R.2**. I would encourage the Authors to use either SIE or SIC in the Introduction, the current version has both of them. Also, in Figure 1, I do not see any gray lines, which would refer to the 2000-2020 SIC climatology. Maybe it would aid the interpretation of Fig. 1 if it had multiple panels instead of the contour lines. The authors might consider plotting the SIC climatologies with shading in Fig 1 b for example.

Response: Indeed, sea ice extent (SIE) and sea ice concentration (SIC) are two similar parameters describing the areal coverage of sea ice, while the former denotes the "boundary" and the latter

represents the "spatial fraction". As suggested, we will use SIE in the Introduction. SIC will be used only once when showing the spatial pattern of sea ice cover anomalies to detect the regions where severe ice loss occurred in July 2020 (Fig. 1.). Besides, Fig. 1 has been modified to aid the interpretation. We plot spatial patterns of SIC anomalies and the SIEs in different panels instead of superimposing the contour lines onto the shading (figure attached below).

[Figure]

**Figure 1.** (a) Spatial patterns of SIC anomalies (shading), and (b) the SIEs in typical years (bold lines). The red line represents the SIE in July 2020. Green and grey curves within denote the SIE in July 2012 and the 20-yr average of the recent period 2001-2020, respectively. The anomalies are computed as the difference between the fields in July and the corresponding climatology over the past four decades (1979-2020). Pink polygons encapsulate areas where substantial sea ice cover loss (60° E-165° E, 70° N-82° N) occurred in July 2020, which represents the study area of this paper.

**R.3.** In general, in the figure captions it would be helpful not to use abbreviations.

Response: We will add the full names of the abbreviations in the figure captions.

**R.4.** In many cases, the significance of the anomalies are not clear. In Fig 2, Fig. 4 and Fig.5 it would be necessary to include significance as stippling for the anomalies. In Fig. 6, I do not see the significance of the results (nor statistically or literally). For example, in lines 237-240, the energy convergence should start early March and peak in June in each year corresponding with solar irradiation seasonality. How are the results presented in Fig 6 differ from the climatology? e.g., a histogram of all 42 years' melt start date could help to point out that 2020 May melt start was statistically significantly earlier than usual. Polishing the discussion of Fig. 6 would be essential to help the reader arrive at the conclusions that the authors set forth.

Response: Thanks for the insightful comments on the significance test. The significance of the

figures and results is an essential issue when drawing a conclusion. Following the suggestion, we have added striplings to denote anomalies that are significant (e.g. greater than two standard deviations) in Fig 2, Fig. 4, and Fig.5. Accordingly, we will polish the discussion part of these figures (Fig 2, Fig. 4, and Fig.5) for better clarification. The significance of the results shown in Fig.6, including the magnitudes of different anomalies will be stated literally in the paragraph of its analysis. Besides, we produce a bar plot of all 42 years' early melt date to distinguish the particularly early melt onset in 2020. The revised figures are shown below.

[Figure]

**Figure 2.** Spatial patterns of sea level pressure anomalies (shading) during April to June 2020. The anomalies are computed as the difference between the averaged fields of the three months (April-June) and the corresponding climatology over the past four decades (1979-2020). Stipplings represent the values where the anomaly exceeds 1.5 standard deviations.

[Figure]

**Figure 4.** Vertical cross-sections of zonal averaged (a) air temperature and (b) specific humidity anomalies, as a function of latitude and pressure level, during the spring months (April–June) of 2020 spanning the regions with significant energy and moisture convergence (60° E-165° E, 60° N-90° N). The anomalies are calculated as the difference between the averaged fields of the three months (April-June) and the corresponding climatology over the past four decades (1979-2020). Stipplings represent the values where the anomaly exceeds 2 standard deviations.

[Figure]

**Figure 5.** Anomalies of surface (a) downwelling and (b) net longwave radiation, (d) downwelling and (e) net shortwave radiation, as well as sensible (c) and latent (f) heat fluxes. The anomalies are relative to the climatology

with monthly resolution from the years 1979-2020 and averaged over the spring months (April–June) of 2020. The stippled grids denote those with values where the anomaly exceeds 2 standard deviations.

[Figure]

**Figure 6.** (a) Time series of sea ice extent, the anomalies of atmospheric energy transport convergence and surface energy fluxes over the study area (indicated by the green polygon in Fig. 3c and d) during 2020. The blue curve represents the SIE. The red line denotes the anomalies of net solar radiation. The green line corresponds to the anomalies of the sum of the downwelling thermal radiation and the turbulent (latent plus sensible) flux. The vertical pink line denotes the average melt day (May 28) in 2020, provided by NASA. The anomalies are relative to the climatology of the years 1979-2020. (b) The averaged melt date of the study area during the period 1979-2020. The grey dashed line represents the mean melt date of these four decades. Error bars denote one standard deviation.

L264: significant is what sense? If statistically, please provide the p value.

Also, when stating 99% significance, what was the applied significance testing method?

Response: The decreasing trend detected in the averaged sea ice thickness of the study area in

spring during the period 1979-2020 is significant at the 99% confidence level, which is specified

at Line 267 in the original manuscript, using a Student's t-test. The significance testing method

will be added to the revised paper.

**R.4.** I think a more thorough discussion of Fig.10a would also improve the paper. Any hints on the

seen low-frequency oscillation in the 10-yr trends? Can this be linked with large-scale circulation

trends (not SLP, but winds or upper-level geopotential, e.g., 300hPa)?

Response: Intuitively, the low-frequency oscillation in the 10-yr trends will be closely tied with

the large-scale circulation trends as the large-scale circulation plays a vital role in regulating

cyclones and moisture/total energy flux. We will do some work to look for the upper-level

large-scale circulation and try to find out the connection. In the revised paper, we will discuss

more thoroughly about Fig.10a.

**Reference:**

Baxter, I., Ding, Q., Schweiger, A., L'Heureux, M., Baxter, S., Wang, T., Zhang, Q., Harnos, K., Markle, B., and Topal, D.: How tropical Pacific surface cooling contributed to accelerated sea ice melt from 2007 to 2012 as ice is thinned by anthropogenic forcing, Journal of Climate, 32, 8583-8602, 2019.

Bi, H., Wang, Y., Liang, Y., Sun, W., Liang, X., Yu, Q., Zhang, Z., and Xu, X.: Influences of Summertime Arctic Dipole Atmospheric Circulation on Sea Ice Concentration Variations in the Pacific Sector of the Arctic during Different Pacific Decadal Oscillation Phases Journal of Climate, 34, 3003-3019, 10.1175/jcli-d-19-0843.1, 2021.

Ding, Q., Schweiger, A., L'Heureux, M., Battisti, David S., Po-Chedley, S., Johnson, Nathaniel C., Blanchard-Wrigglesworth, E., Harnos, K., Zhang, Q., Eastman, R., and Steig, Eric J.: Influence of high-latitude atmospheric circulation changes on summertime Arctic sea ice, Nature Climate Change, 7, 289-295, 10.1038/nclimate3241, 2017.

Graversen, R. G., Mauritsen, T., Drijfhout, S., Tjernstrom, M., and Martensson, S.: Warm winds from the Pacific caused extensive Arctic sea-ice melt in summer 2007, Climate Dynamics, 36, 2103-2112, 2011.

Horvath, S., Stroeve, J., Rajagopalan, B., and Jahn, A.: Arctic sea ice melt onset favored by an atmospheric pressure pattern reminiscent of the North American-Eurasian Arctic pattern, Climate Dynamics, 57, 1771-1787, 10.1007/s00382-021-05776-y, 2021.

Kapsch, M.-L., Skific, N., Graversen, R. G., Tjernström, M., and Francis, J. A.: Summers with low Arctic sea ice linked to persistence of spring atmospheric circulation patterns, Climate Dynamics, 52, 2497-2512, 10.1007/s00382-018-4279-z, 2019.

Ogi, M., and Wallace, J. M.: The role of summer surface wind anomalies in the summer Arctic sea ice extent in 2010 and 2011, Geophysical Research Letters, 39,

https://doi.org/10.1029/2012GL051330, 2012.

Screen, J. A., and Deser, C.: Pacific Ocean Variability Influences the Time of Emergence of a Seasonally Ice-Free Arctic Ocean, Geophysical Research Letters, 46, 2222-2231, https://doi.org/10.1029/2018GL081393, 2019.

Vázquez, M., Nieto, R., Drumond, A., and Gimeno, L.: Extreme Sea Ice Loss over the Arctic: An Analysis Based on Anomalous Moisture Transport, Atmosphere, 8, 10.3390/atmos8020032, 2017.

Warner, J., Screen, J., and Scaife, A.: Links between Barents‑Kara sea ice and the extratropical atmospheric circulation explained by internal variability and tropical forcing, Geophysical Research Letters, 47, e2019GL085679, 2020.

---

## Author Response (AR1)

**Response to Referee #1**

Review of the Manuscript 'Warm and moist atmospheric flow caused a record minimum July sea ice extent of the Arctic in 2020' by Ling et al. submitted to The Cryosphere.

**Summary:**

The authors are exploring the atmospheric conditions during spring that might have led to the low sea-ice extent in July of 2020. In their analysis the authors focus on the transport and convergence of moist and warm air masses and associated changes in the surface energy balance. Using a cyclone tracking algorithm, they connect the increased energy transport in spring 2020 to anomalies in the cyclone activity. Thereby, the study follows up on a range of previous studies, which identified the spring atmospheric conditions to be the major driver of a low summer sea-ice extent. While the topic is very relevant and interesting, the analysis lacks explanations and potentially also extensions.

**General comments:**

The analysis is rather comprehensive but the methods and supporting information are not always clear, hence, it is hard to arrive at the drawn conclusions. One of the main problems is that the study area contains a lot of land points, but the focus of interest is sea-ice variability. Why did the authors choose this study area and did not e.g. exclude land points or even focus on the area that showed the largest SIE anomalies in 2020 from Fig. 1.? Another point is the cyclone detection and conclusions drawn. It is not clear how robust the results are.

Response:

a. First of all, thanks a lot for the advice on this manuscript which helps us to improve the research. In the present study, we use a range of latitudes and longitudes to define a relatively regular study area (60° E-165° E, 70° N-82° N) for the convenience of plotting. However, in the analysis, the retrieved values are averaged over the oceanic grids by applying a land mask and SIE is defined as areas that have an ice concentration of at least 15% (Fig. 6 and 7).

b. The nature that extratropical cyclones are characterized by great complexity (asymmetric

structure, differ rather more in size, multi-centers and occur in very diverse synoptic situations) indicate that there is no single commonly agreed upon scientific definition of an extratropical cyclone, and there exists a range of ideas and concepts regarding how to identify and track them (Murray and Simmonds, 1991; Serreze et al., 1993; Serreze, 1994; Sinclair, 1994; Pinto et al., 2005; Wang et al., 2006; Wernli and Schwierz, 2006). Methods differ in a number of aspects including variables used, tracking parameters, and post-processing. Different approaches each have their strengths and weaknesses, hence one cannot "judge" the algorithms to be "right" or "incorrect" (Neu et al., 2013). In the present study, we use a revised automatic cyclone identification and tracking algorithm developed originally by Serreze et al. (1993) to diagnose and track the cyclones from the 6-hourly mean sea level pressure (MSLP) data (Serreze et al., 1993; Serreze, 1995; Serreze et al., 1997; Wu et al., 2006a; Wang et al., 2013). The method was used by the National Oceanic and Atmospheric Administration–Cooperative Institute for Research in Environmental Sciences (NOAA–CIRES) Climate Diagnostics Center (CDC) to diagnose storm tracks for the period 1948 and 2004. Besides, Neu et al. (2013) conducted an intercomparison experiment involving 15 commonly used detection and tracking algorithms for extratropical cyclones. The results revealed that cyclone characteristics that are robust between different schemes and our algorithm agrees well with the others in terms of spatial distribution, interannual variability, and geographical linear patterns of the cyclones. To some extent, these facts give credence to the method utilized in this study.

**Specific comments:**

1) Figure 1: It is not possible to see the colored line indicating the July SIE of 2021 (red) and not possible to distinguish between the others (green, gray). Please choose different colors or a thicker linewidth, as this figure is important for the following analysis.

Response: Fig. 1 has been modified to aid the interpretation. We plot spatial patterns of SIC anomalies and the SIEs in different panels instead of superimposing the contour lines onto the shading. Besides, the study area is also outlined in Fig.1 (Fig. 1 in the revised manuscript).

2) Section 2.2.3: Crawford et al, 2021 ( https://doi.org/10.1175/mwr-d-20-0417.1) have investigated the dependence of spatial and temporal resolution on a realistic detection of cyclone

tracks in ERA-5. How does your algorithm differ from theirs? Do you experience an unrealistic break up of cyclones? The cyclone tracks in Fig. 9 are hard to identify and many end up over land (while you are interested in what happens over the ice), which makes me wonder how robust your whole analysis on the cyclone tracks is. Maybe backwards trajectories would be easier to interpret?

Response:

a. Crawford et al. (2021) aimed to test the sensitivity of the cyclone tracking method to the spatial and temporal resolution of ERA5 sea level pressure fields. The cyclone detection and tracking algorithm they used was introduced by Crawford and Serreze (2016) and builds on the method originally designed by Serreze et al. (1993). Coincidentally, we use a revised automatic cyclone identification and tracking algorithm which is based on the same scheme developed by Serreze et al. (1993). The main difference is that they explicitly identify multicenter cyclones (a single cyclone may contain two or more distinct but closely related minima in SLP) as well as splitting and merging events. While in our algorithm, the exact center is determined as the grid with the largest local Laplacian of SLP when multiple cyclone center candidates are found within a radius of 600 km.

b. In the present study, the ERA5 6-hour SLP fields were interpolated to a 50-km version of the NSIDC EASE-grid, prior to the application of the algorithm. As suggested by Crawford et al. (2021), we used a common search distance (7×7 array of grid points) when detecting minima in sea level pressure to avoid unrealistic break up of cyclones.

c. Some studies corroborated the fact that synoptic cyclones play a crucial role in regulating the poleward moisture and energy fluxes (Jakobson and Vihma, 2010; Dufour et al., 2016; Villamil-Otero et al., 2018). As a cyclone represents a dynamical process and concerning its fundamental nature in holding moisture and energy, all poleward cyclones may play a non-negligible role in transporting energy and water vapor to the Arctic in the form of a relay. Thus, rather than just confining cyclones that occurred over the ice, we take all northward cyclones into account when inspecting the underlying effects of the cyclones on the meridional transport of energy/moisture in spring 2020. As a consequence, some cyclones in Fig.9 had their tracks end over land. Tracking cyclones backwards from their lysis to form

trajectories may be more straightforward in this study. However, to our knowledge, there exist no backward-tracking schemes of an extratropical cyclone. It can be an innovative path for future research.

3) Line 157: A low-pressure anomaly over the central Arctic dominates the spring of 2020. Similar anomalies were detected in spring of years with a low summer sea ice in Kapsch et al., 2019 (https://doi.org/10.1007/s00382-018-4279-z) and Horvath et al., 2021 (https://doi.org/10.1007/s00382-021-05776-y). Both of the studies pointed out that a similar pattern was associated with summers of low sea ice and an early melt onset in the Kara/East Siberian Sea. You should relate to these studies, as your findings for 2020 are a confirmation their findings.

Response: We have read the recommended literature and related these two studies in our study when discussing the atmospheric pattern in spring 2020 (Line 182-185 in the revised manuscript).

4) Fig. 6: the total convergence of energy is heavily smoothed. Why using a different temporal resolution for the different variables? Please clarify. A higher spatial resolution can also give an idea about the persistence of atmospheric circulation patterns that lead to the enhanced energy transport, which was found to be of importance for the summer sea ice in previous studies.

Response: We have no idea why the hourly REA5 convergence of total energy flux fields are rather noisy (Figure below, grey dashed line), the reasoning behind it demands further evaluations. Hence it is a compromise to utilize the monthly mean of the convergence fields.

[Figure]

**Figure 6.** Time series of SIE, the anomalies of atmospheric energy transport convergence and surface energy

fluxes over the study area (indicated by the green polygon in Fig. 3c and d) during 2020. The blue curve represents the SIE. The red line denotes the anomalies of net solar radiation. The green line corresponds to the anomalies of the sum of the downwelling thermal radiation and the turbulent (latent plus sensible) flux. The vertical pink line denotes the average melt day (May 28) in 2020, provided by NASA. The anomalies are relative to the climatology of the years 1979-2020.

5) Line 46: '… various disciplines.' – like which?  Line 87: Schweiger et al outlined a less than 0.1m difference and a high pattern correlation. How different are the data sets over the area of interest?

Response:

a.  The scientific studies about the causes of Arctic sea ice shrinkage encompass various disciplines, including atmospheric (Deser et al., 2000; Wu et al., 2006b; Wang et al., 2009; Ogi et al., 2016) and oceanic (Årthun et al., 2012; Miles et al., 2014; Zhang, 2015; Årthun and Eldevik, 2016) sciences. This paper aimed to assess the impact of the variations in atmospheric transport of total energy and moisture on sea ice loss in spring 2020, the second paragraph in the introduction, therefore, discusses the current understanding of relevant mechanisms (Line 38-54 in the revised manuscript).

b.  In the Laptev and East Siberian Seas, ICESat and PIOMAS ice thickness fields show a close agreement with the spatial pattern of ice thickness. The PIOMAS thickness fields are about 0.2-0.7 m small than that of ICESat for February–March (Schweiger et al., 2011). The description of the PIOMAS fields in the study area has been added in the revised version (Line 104-106 in the revised manuscript).

6) Line 119: You claim that the results of your energy flux estimates are similar to those of ERA-5. If the moisture flux exists in ERA-5, why estimating it?

Response: Indeed, the ERA5 fields of the total energy and moisture flux experienced an update and some corrections during the period we processed the data. To ensure that our research continues, we calculate the moisture flux when the corresponding field from ERA5 was in an upgrade state. Once the ERA5 field is prepared, we compared our estimated results with ERA5 and decided to directly utilize the ERA5 fields of the total energy and moisture flux.

7) Fig. 5: Might be an optical illusion due to the projection, but for me it seems that the study area slightly differs from the one indicated in Fig. 3. It seems that there are more land points in Fig. 5.

However, see comment on excluding land points from the analysis.

Response: All the fields obtained from ERA5 have a spatial resolution of 1.0° ×1.0° in longitude and latitude. For better illustration of the Arctic region, the anomalies of different meteorological variables as well as cyclone characteristics are displayed on a polar stereographic projection (Figs. 1, 2, 3, 5, 7, 10, and 11). The spatial patterns of variations of surface radiative and turbulent fluxes shown in Fig.5 are not the optical illusion but indeed a true phenomenon exists in the data. The figure attached below shows the same anomalies, which have coincident patterns with Fig.5. Besides, the plotted view of Fig.5 is slightly larger than the study area in order to display (has been clarified in the caption of Fig.5). In the analysis, the retrieved values are averaged over the oceanic grids within the study area by applying a land mask and SIE is defined as areas that have an ice concentration of at least 15% (Fig. 6 and 7).

[Figure]

**Figure 5.** Anomalies of surface (a) downwelling and (b) net longwave radiation, (d) downwelling and (e) net shortwave radiation, as well as sensible (c) and latent (f) heat fluxes. The anomalies are relative to the climatology with monthly resolution from the years 1979-2020 and averaged over the spring months (April–June) of 2020. The stippled grids denote those with values where the anomaly exceeds 2 standard deviations.

8) Line 284: I don't see how calculating the cyclones from ERA-Interim gives more credibility in the methods and results. It might be worse to take a more independent reanalysis for such credibility check. Again, a discussion on the method and previous findings is necessary (see point Section 2.2.3).

Response: We sincerely appreciate the valuable comments on the credence check of the cyclone identifying and tracking algorithm. In a previous study of our team, the retrieved cyclones from ERA-Interim SLP fields are used to discuss the impact of cyclones on the sea ice area flux through the Baffin Bay with respect to dynamical processes (Liang et al., 2021). For convenience, we compared the cyclone systems diagnosed from the ERA5 SLP with those from ERA-Interim,

which may be problematic. We have deleted these sentences in the revised manuscript and referred to the results of the intercomparison between 15 different algorithms in Neu et al. (2013) (Line 322-326 in the revised manuscript).

9) Line 361: Ice motion in response to the circulation patterns and cyclones should be discussed a bit more in detail, as it is an important process. It also should be related to previous studies. There have also been other studies, elaborating on some of the processes that lead to an earlier melt onset (e.g. increased liquid precipitation).

Response: The main driver of sea ice motion is the surface wind, which can explain more than 70% of the variance of the ice velocity in the central Arctic Ocean. Sea ice tends to move with a speed of about 2% of the surface wind and about 45° to the right of the wind (Thorndike and Colony, 1982). That is to say, variations in large-scale circulation patterns and cyclones would inevitably change the ice drift pattern. We discussed the ice motion and other dynamical processes in more detail in the revised paper and add the related references (Line 420-436 in the revised manuscript). Besides, the literature elaborating the consistent mechanisms with the present study which lead to an earlier melt onset has added to the discussion part of Fig.6 (Line 282-285 in the revised manuscript).

10) Line 415: A very relevant study related to an early melt onset in years of low summer sea ice in the study area is also Mortin et al., 2016 (https://doi.org/10.1002/2016GL069330) as well as several studies by Stroeve et al.

Response: We have read the recommended literature and referred to them when discussing Fig.6 in our study (Line 282-285 in the revised manuscript).

11) Line 427: It should be mentioned much earlier, probably in the introduction, that the September SIE was not a record in 2021. It might be interesting for the reader to know why this study explores the July SIE instead of the September SIE.

Response: We clarified the fact that the September SIE of 2021 did not hit the record earlier in the introduction (Line 55-56 in the manuscript). In 2020 however, Arctic sea ice experienced the lowest July extent recorded since 1979, which is ~21% lower than the average July SIE over the period 2000-2020. This study aims to disentangle the mechanisms that drive this extreme event. For extension, we added a paragraph to discuss the connection between July and September Arctic

sea ice extent in 2020 (Line 480-493 in the revised manuscript).

**Technical corrections:**

1) Line 102: 'replacing ERA-Interim'

Response: We replaced the sentence with "ERA5 represents a new reanalysis product which improves on its predecessor (ERA-Interim). It benefits from a decade of developments in model physics, core dynamics, and data assimilation." (Line 119-120 in the revised manuscript).

2) Line 175: remove parenthesis behind Kara Sea.

Response: Revised as suggested.

3) Line 188: 'unusual conditions with higher'

Response: Revised as suggested (Line 213 in the revised manuscript).

4) Fig. 1, 2, 3, 7, 10, 11: It might be worse to use one projection (including latitude range) for the plots.

Response: As the regions of interest in this study are located in the Arctic, I think it is better to plot the geographical patterns of different variables on a polar stereographic projection. Moreover, all figures that demonstrate spatial distributions have the same projection for the sake of uniformity of presentation.

5) Fig. 4: Line 200: 'spanning the with significant' – something missing here. The whole caption would benefit from a revision.

Response: The expression has been changed into "spanning the regions with significant" (Line 226 in the revised manuscript).

6) Line 282: 'we identify and track cyclone'

Response: Corrected as suggested (Line 320 in the revised manuscript).

7) In many places there is no space between text and the following parentheses, e.g. Line 205, 256, 279, 283, 398, 419, 421 … In general, it would be good to check for spelling and language related issues.

Response: Following the suggestion, the space between text and the following parentheses have been added. We also further polished the revised manuscript and remove the inappropriate expressions.

**Response to Referee #2**

Review of the Manuscript 'Warm and moist atmospheric flow caused a record minimum July sea ice extent of the Arctic in 2020' by Ling et al. submitted to The Cryosphere.

**Summary:**

Liang et al. aims to investigate the July 2020 extreme sea ice melt event in terms of physical mechanisms. They look at the prior late spring-early summer 2020 to explain that anomalous warm air intrusion and cyclone activity set up favorable conditions for sea ice melt in July 2020. I find the idea interesting and well suited for The Cryosphere journal and the methods generally appear sound, however the presentation of their results and the significance of the findings need a bit more elaboration before I could recommend the paper for publication.

**Reviewer comments**

**R.1.** I find the Introduction a bit hard to follow. The authors might consider reorganizing it a little bit via discussing the contents of the current second paragraph before starting to talk about the 2020 SIE extent and referring to Figure 1. From row 30 it reads like it is already the description of the Results. I understand the reasoning behind it; the authors want a succinct Introduction to go with their very specific and well-defined goal in the paper, however I think they could do better in setting up the research question.

Especially, I suggest that the authors discuss more thoroughly the current understanding of oceanic and atmospheric drivers of summer sea ice melt, especially the physical mechanisms, as their objective in this paper is to reveal the underlying mechanisms leading to the record melt in July 2020. For example, in the current introduction the authors only mention surface wind driven sea ice drift as dynamical forcing on sea ice, however in recent years anticyclonic circulation anomalies caused vertical motion (warming and moistening descending air) is also a key component of atmospheric forcing on sea ice (see e.g., Ding et al. 2019; Topal et al. 2020). This local atmosphere-sea-ice coupling mechanism is further linked to large-scale circulation changes and forcing from the tropics especially over the enhanced melt period between 2007 and 2012 (Screen and Deser 2020; Warner et al. 2020; Baxter et al. 2019). Therefore, the well-known

thermodynamical factors causing sea ice melt may be better linked with known dynamical sources besides surface wind drift, which is far from being the only dynamics causing sea ice variations in the Arctic. In this way the authors may set up their research question a bit more connected to existing literature and highlight that their goal is to complement the existing knowledge of dynamical drivers of sea ice loss which can well be exemplified via a case study in July 2020.

June-August 2020 was dominated by a high-pressure anomaly in the Arctic, which could have acted in concert with the prevailing spring conditions to cause the sea ice extreme melt. I wonder if the authors could provide more discussion on how they distinguish their results or link together with previous literature either in the Introduction or in their Discussion part.

Response: We sincerely appreciate the constructive and detailed comments by Referee #2. These comments helped us improve our manuscript, and provided important guidance for our future research.

a. The Introduction has been reorganized as suggested. Before analyzing the minimum sea ice extent in July 2020, we discussed thoroughly the current understanding of atmospheric drivers of sea ice melt, especially the relevant physical mechanisms and refer to the previous studies. Then we present the extreme event of sea ice loss in July 2020, and the scientific question of the present research is set up afterward (Line 29-77 in the revised manuscript).

b. Although the September SIE of 2020 did not shatter the previous lows to be a new record, September 2020 had the second-lowest SIE since 1979, stood at $3.82\times10^6$ km$^2$, which is merely 12% higher than the lowest SIE. A prominent high-pressure anomaly dominated the Arctic in July-September 2020 (especially in July-August, the figure below). Previous studies elaborated that the recent summertime sea ice depletion is broadly associated with the anticyclonic atmospheric circulation pattern which can increase the downwelling longwave radiation above the ice by warming and moistening the lower troposphere (Ogi and Wallace, 2012; Ding et al., 2017). The combination of low-pressure anomaly persistent in April-June (favoring moisture and energy inflow) and anticyclonic atmospheric circulation pattern (leading to adiabatic warming) may contribute to the particularly low SIE of September 2020, the mechanisms of which would be the potential candidates for future studies. The present study is dedicated to elucidating that anomalous high inflow of total energy and moisture from lower latitudes to the Arctic in spring caused severe sea ice loss of July 2020. The above

arguments about the anticyclonic atmospheric circulation pattern, therefore, have been added to the Discussion and Conclusions part when mentioning the September SIE of 2020 (Line 480-493 in the revised manuscript).

[Figure]

**Supplementary Figure.** Spatial patterns of sea level pressure anomalies (shading) during July to August 2020. The anomalies are computed as the difference between the averaged fields of the three months (April-June) and the corresponding climatology over the past four decades (1979-2020). Stipplings represent the values where the anomaly exceeds 1.5 standard deviations.

c.   Several existing literature pointed out similar mechanisms with ours (Graversen et al., 2011; Vázquez et al., 2017; Kapsch et al., 2019; Horvath et al., 2021). Our results serve to augment more evidence to the mechanisms that drive sea ice loss through transporting moisture and energy into the Arctic via a case study in July 2020. Here we argued that the unusual atmospheric energy and moisture transport favored by large-scale circulation and cyclones in Spring 2020 effectively reduce ice extent under the circumstance of more thin, first-year ice, which is a novel result. These points distinguishing our research from previous literature has been declared in the revised version (Line 473-475 in the revised manuscript).

**R.2**. I would encourage the Authors to use either SIE or SIC in the Introduction, the current version has both of them. Also, in Figure 1, I do not see any gray lines, which would refer to the 2000-2020 SIC climatology. Maybe it would aid the interpretation of Fig. 1 if it had multiple panels instead of the contour lines. The authors might consider plotting the SIC climatologies with shading in Fig 1 b for example.

Response: Indeed, sea ice extent (SIE) and sea ice concentration (SIC) are two similar parameters describing the areal coverage of sea ice, while the former denotes the "boundary" and the latter

represents the "spatial fraction". As suggested, we use SIE in the Introduction. SIC is used only once when showing the spatial pattern of sea ice cover anomalies to detect the regions where severe ice loss occurred in July 2020 (Fig. 1.). Besides, Fig. 1 has been modified to aid the interpretation following the suggestion. We plot spatial patterns of SIC anomalies and the SIEs in different panels instead of superimposing the contour lines onto the shading.

**R.3.** In general, in the figure captions it would be helpful not to use abbreviations.

Response: We added the full names of the abbreviations in the figure captions.

**R.4.** In many cases, the significance of the anomalies are not clear. In Fig 2, Fig. 4 and Fig.5 it would be necessary to include significance as stippling for the anomalies. In Fig. 6, I do not see the significance of the results (nor statistically or literally). For example, in lines 237-240, the energy convergence should start early March and peak in June in each year corresponding with solar irradiation seasonality. How are the results presented in Fig 6 differ from the climatology? e.g., a histogram of all 42 years' melt start date could help to point out that 2020 May melt start was statistically significantly earlier than usual. Polishing the discussion of Fig. 6 would be essential to help the reader arrive at the conclusions that the authors set forth.

Response: Thanks for the insightful comments on the significance test. The significance of the figures and results is an essential issue when drawing a conclusion. Following the suggestion, we have added striplings to denote anomalies that are significant (e.g. greater than two standard deviations) in Fig 2, Fig. 4, and Fig.5. Accordingly, we polished the discussion part of these figures (Fig 2, Fig. 4, and Fig.5) for better clarification. The significance of the results shown in Fig.6, including the magnitudes of different anomalies have been stated literally in the paragraph of its analysis (Line 266-285 in the revised manuscript). Besides, we produce a bar plot of all 42 years' early melt date to distinguish the particularly early melt onset in 2020. The revised figures are replaced in the revised version.

L264: significant is what sense? If statistically, please provide the p value.

Also, when stating 99% significance, what was the applied significance testing method?

Response: The decreasing trend detected in the averaged sea ice thickness of the study area in spring during the period 1979-2020 is significant at the 99% confidence level, using a Student's t-test. The significance testing method has been clarified in the revised paper clarified (Line 304 in the revised manuscript).

**R.4.** I think a more thorough discussion of Fig.10a would also improve the paper. Any hints on the seen low-frequency oscillation in the 10-yr trends? Can this be linked with large-scale circulation trends (not SLP, but winds or upper-level geopotential, e.g., 300hPa)?

Response: By processing the data, we found that the low-frequency variations in the 10-yr trends are controlled by the large-scale circulation trends. A more thorough discussion of Fig.10a has been presented (Line 383-390 in the revised manuscript). However, this question indeed requires more comprehensive analysis and should be explored in greater detail in future studies.

Årthun, M., Eldevik, T., Smedsrud, L., Skagseth, Ø., and Ingvaldsen, R.: Quantifying the influence of Atlantic heat on Barents Sea ice variability and retreat, Journal of Climate, 25, 4736-4743, 2012.

Årthun, M., and Eldevik, T.: On anomalous ocean heat transport toward the Arctic and associated climate predictability, Journal of Climate, 29, 689-704, 2016.

Crawford, A. D., and Serreze, M. C.: Does the Summer Arctic Frontal Zone Influence Arctic Ocean Cyclone Activity?, Journal of Climate, 29, 4977-4993, 10.1175/jcli-d-15-0755.1, 2016.

Crawford, A. D., Schreiber, E. A. P., Sommer, N., Serreze, M. C., Stroeve, J. C., and Barber, D. G.: Sensitivity of Northern Hemisphere Cyclone Detection and Tracking Results to Fine Spatial and Temporal Resolution Using ERA5, Monthly Weather Review, 149, 2581-2598, 10.1175/mwr-d-20-0417.1, 2021.

Deser, C., Walsh, J. E., and Timlin, M. S.: Arctic sea ice variability in the context of recent atmospheric circulation trends, Journal of Climate, 13, 617-633, Doi 10.1175/1520-0442(2000)013<0617:Asivit>2.0.Co;2, 2000.

Ding, Q., Schweiger, A., L'Heureux, M., Battisti, David S., Po-Chedley, S., Johnson, Nathaniel C., Blanchard-Wrigglesworth, E., Harnos, K., Zhang, Q., Eastman, R., and Steig, Eric J.: Influence of high-latitude atmospheric circulation changes on summertime Arctic sea ice, Nature Climate Change, 7, 289-295, 10.1038/nclimate3241, 2017.

Dufour, A., Zolina, O., and Gulev, S. K.: Atmospheric moisture transport to the Arctic: Assessment of reanalyses and analysis of transport components, Journal of Climate, 29, 5061-5081, 2016.

Graversen, R. G., Mauritsen, T., Drijfhout, S., Tjernstrom, M., and Martensson, S.: Warm winds from the Pacific caused extensive Arctic sea-ice melt in summer 2007, Climate Dynamics, 36, 2103-2112, 2011.

Horvath, S., Stroeve, J., Rajagopalan, B., and Jahn, A.: Arctic sea ice melt onset favored by an atmospheric pressure pattern reminiscent of the North American-Eurasian Arctic pattern, Climate Dynamics, 57, 1771-1787, 10.1007/s00382-021-05776-y, 2021.

Jakobson, E., and Vihma, T.: Atmospheric moisture budget in the Arctic based on the ERA‐40 reanalysis, International Journal of Climatology, 30, 2175-2194, 2010.

Kapsch, M.-L., Skific, N., Graversen, R. G., Tjernström, M., and Francis, J. A.: Summers with low Arctic sea ice linked to persistence of spring atmospheric circulation patterns, Climate Dynamics, 52, 2497-2512, 10.1007/s00382-018-4279-z, 2019.

Liang, Y., Bi, H., Wang, Y., Huang, H., Zhang, Z., Huang, J., and Liu, Y.: Role of Extratropical Wintertime Cyclones in Regulating the Variations of Baffin Bay Sea Ice Export, Journal of Geophysical Research: Atmospheres, 126, e2020JD033616, 2021.

Miles, M. W., Divine, D. V., Furevik, T., Jansen, E., Moros, M., and Ogilvie, A. E.: A signal of persistent Atlantic multidecadal variability in Arctic sea ice, Geophysical Research Letters, 41, 463-469, 2014.

Murray, R. J., and Simmonds, I. H.: A numerical scheme for tracking cyclone centres from digital data. Part II: application to January and July general circulation model simulations, Australian Meteorological Magazine, 39, 167-180, 1991.

Neu, U., Akperov, M. G., Bellenbaum, N., Benestad, R., Blender, R., Caballero, R., Cocozza, A., Dacre, H. F., Feng, Y., and Fraedrich, K.: IMILAST: A community effort to intercompare extratropical cyclone detection and tracking algorithms, Bulletin of the American Meteorological Society, 94, 529-547, 2013.

Ogi, M., and Wallace, J. M.: The role of summer surface wind anomalies in the summer Arctic sea ice extent in 2010 and 2011, Geophysical Research Letters, 39, https://doi.org/10.1029/2012GL051330, 2012.

Ogi, M., Rysgaard, S., and Barber, D. G.: Importance of combined winter and summer Arctic Oscillation (AO) on September sea ice extent, Environ Res Lett, 11, Artn 034019 10.1088/1748-9326/11/3/034019, 2016.

Pinto, J. G., Spangehl, T., Ulbrich, U., and Speth, P.: Sensitivities of a cyclone detection and tracking algorithm: individual tracks and climatology, Meteorologische Zeitschrift, 14, 823-838, 2005.

Schweiger, A., Lindsay, R., Zhang, J., Steele, M., Stern, H., and Kwok, R.: Uncertainty in modeled Arctic sea ice volume, 116, https://doi.org/10.1029/2011JC007084, 2011.

Serreze, M. C., Box, J. E., Barry, R. G., and Walsh, J. E.: Characteristics of Arctic synoptic activity, 1952–1989, Meteorology and Atmospheric Physics, 51, 147-164, 1993.

Serreze, M. C.: Climatological aspects of cyclone development and decay in the Arctic, Journal of Atmosphere, 33, 1-23, 1994.

Serreze, M. C.: Climatological aspects of cyclone development and decay in the Arctic, Journal of Atmosphere, 33, 1-23, 1995.

Serreze, M. C., Carse, F., Barry, R. G., and Rogers, J. C.: Icelandic low cyclone activity: Climatological features, linkages with the NAG, and relationships with recent changes in the Northern Hemisphere circulation, Journal of Climate, 10, 453-464, Doi 10.1175/1520-0442(1997)010<0453:Ilcacf>2.0.Co;2, 1997.

Sinclair, M. R.: An objective cyclone climatology for the Southern Hemisphere, Monthly Weather Review, 122, 2239-2256, 1994.

Thorndike, A., and Colony, R.: Sea ice motion in response to geostrophic winds, Journal of Geophysical Research: Oceans, 87, 5845-5852, 1982.

Vázquez, M., Nieto, R., Drumond, A., and Gimeno, L.: Extreme Sea Ice Loss over the Arctic: An Analysis Based on Anomalous Moisture Transport, Atmosphere, 8, 10.3390/atmos8020032, 2017.

Villamil-Otero, G. A., Zhang, J., He, J., and Zhang, X.: Role of extratropical cyclones in the recently observed increase in poleward moisture transport into the Arctic Ocean, Advances in Atmospheric Sciences, 35, 85-94, 2018.

Wang, J., Zhang, J., Watanabe, E., Ikeda, M., Mizobata, K., Walsh, J. E., Bai, X., and Wu, B.: Is the Dipole Anomaly a major driver to record lows in Arctic summer sea ice extent?, Geophysical Research Letters, 36, 10.1029/2008gl036706, 2009.

Wang, X. L., Swail, V. R., and Zwiers, F. W.: Climatology and changes of extratropical cyclone activity: Comparison of ERA-40 with NCEP–NCAR reanalysis for 1958–2001, Journal of Climate, 19, 3145-3166, 2006.

Wang, X. L., Feng, Y., Compo, G., Swail, V., Zwiers, F., Allan, R., and Sardeshmukh, P.: Trends and low frequency variability of extra-tropical cyclone activity in the ensemble of twentieth century reanalysis, Climate Dynamics, 40, 2775-2800, 2013.

Wernli, H., and Schwierz, C.: Surface cyclones in the ERA-40 dataset (1958–2001). Part I: Novel identification method and global climatology, Journal of the atmospheric sciences, 63, 2486-2507, 2006.

Wu, B., Wang, J., and Walsh, J. E.: Dipole anomaly in the winter Arctic atmosphere and its association with sea ice motion, Journal of Climate, 19, 210-225, 2006a.

Wu, B. Y., Wang, J., and Walsh, J. E.: Dipole anomaly in the winter Arctic atmosphere and its association with sea ice motion, Journal of Climate, 19, 210-225, Doi 10.1175/Jcli3619.1, 2006b.

Zhang, R.: Mechanisms for low-frequency variability of summer Arctic sea ice extent, Proceedings of the National Academy of Sciences, 112, 4570-4575, 2015.

---

## Author Response (AR2)

**Response to Editor**

**Dear Editor David Schroeder:**

Thank you very much for handling and review of our manuscript. In the latest revised manuscript, we have addressed and answered all the comments and questions raised by the reviewers. Besides, we checked the manuscript carefully for typos and refined some of the descriptions

The revisions associated with the method, analysis, and figures are summarized as follows:

1) Method: Utilizing composite analysis, we conduct maps of the top 5 years with the lowest SIE in the study area and those 5 years with the highest SIE. Based on the result, we address the concern of Referee #2 about the similarities of the atmospheric conditions between spring 2020 and that of previous studies. We also explain the source and deviation of the sea-ice growth model as suggested by Referee #4.

2) Analyses: We have revised the discussion according to the composite analysis. Moreover, we added discussion about the moistening and warming that occurred over land, as well as their potential impacts on the Arctic Ocean system, to address the comments raised by Referee #3. The errors about measurements for clouds are also clarified.

Figures: We have replotted figures using a more commonly used projection with central longitude
 0°. Several details of the figures and the captions have been also improved.

Please find below our point-by-point response (blue text) to the comments (black text) by the reviewers.

**Best regards**

Yu Liang and co-authors

**Response to Referee #2**

**Summary**

Liang et al. addressed my concerns that I have raised in the first round of revision of their manuscript, which I appreciate. However, based on my careful examination of the authors response to Reviewer#1 and the revised manuscript I still have concerns, which need clarifications before I can recommend the paper for publication.

**Major concerns**

R.1. Reviewer #1 pointed out two seemingly relevant papers (Kapsch et al. 2019 and Horvath et al. 2021), however, to me, either of the papers show similar geopotential height/SLP structure as the authors of the present manuscript (their Fig. 1 SLP pattern). Although both papers suggest that low-pressure anomaly in the central Arctic in spring may favor anomalous melt onset, either of the SOM nodes show the particular case in 2020 SLP, as shown in Fig.1 of Liang et al. Some similarities can be recognized looking at node 18 of Fig 3 in Horvath et al. 2021, but similarities solely in the central Arctic with the lower latitudes showing very different SLP structures, which can result in different moisture/energy transports too.

While one single year's seasonal mean (2020 April-May-June) SLP may not be expected to perfectly match any of the nodes in the SOM using daily data, the authors should be more specific in line 182-183 when stating that their results are 'similar' to the SOM analysis of either Kapsch et al. 2019 or Horvath et al. 2021.

Thus, the authors should notice and discuss the potential causes of these differences, which probably can be related to methodological differences. Namely, the authors of the present manuscript use simple anomalies of seasonal mean 2020 spring/early summer. I wonder if the authors will find similar AMJ atmospheric patterns by using a melt-onset-date-based composite of SLP and moisture/energy transport spatial patterns in the Arctic during 1979-2019. Ending with 2019, because adding 2020 to the composite would alter the results, I am curious of. Based on 41 years of data, maybe the difference between those 6 years (approx. 1.5 std dev) with the earliest and those 6 with the latest melt onset (e.g., averaged over the study area) will show similar SLP and moisture/energy transports to the ones seen in 2020 spring/early summer (this similarity may be

quantified using spatial correlation or the significance test of the composite maps based on a twosample t test).

**Response:**

Thanks a lot for the detailed advice on this manuscript. Following the suggestion, we constructed the composite maps of the top 5 years with the lowest SIE in the study area and those 5 years with the highest SIE. The difference of the SLP fields, moisture flux, and total energy flux between them are shown in the figure below (Fig. 1 in the response letter). Note that difference maps of meltonset-date-based composite share similar patterns. Compared to years with higher SIE in the study area, there exists a low-pressure anomaly centered over the study area which extended southwards from the Barents-Kara Seas to the middle part of Eurasia when SIE is low. This low-pressure center is reflected in the SLP anomalies pattern in spring 2020 with a broader extent covering the central Arctic. These kinds of SLP modes favor anomalous high moisture and energy air from Eurasia into the Arctic through the Kara Sea and the Laptev Sea, which agree well with the great transport of moist and warm air mass through the entry. However, in the difference map, the moisture and energy flux partly traveled northward crossing the pole while in spring 2020 the fluxes prevailed in the study area. The discrepancy can be attributed to the significant high-pressure level center with a broad extent in Siberia and the low-pressure center near the pole, indicating that spring 2020 had an unusual atmospheric condition that drove an exceptional form of moisture and energy transport. We carefully checked the SLP pattern of Spring (April to June) 2020 with those SOM nodes in Kapsch et al. (2019) and Horvath et al. (2021). Similarities can be detected between node 18 of Fig 3 in Horvath et al. (2021), with a slight difference in the Bering Sea and the Labrador Sea. Based on the composite analysis, it seems that the difference between atmospheric circulation in 2020 with those SOM nodes is not sensitive to the method used. It underscores the unusual atmospheric condition in spring 2020. We revised the statement when mentioning the similarities in the revised manuscript (Line 188-192, 200 in the revised manuscript).

**Figure 1.** Difference maps of (a) SLP fields, (b) moisture flux, and (c) total energy flux between 5 years with the lowest SIE in the study area and those 5 highest years with the SIE. Stipplings represent the grids with the difference significant at the 90% confidence level (two-sample t-test).

I suggest that it would be safer to say even in the title, that "Warm and moist atmospheric flow" not caused rather either contributed to or preceded "a record minimum July sea ice extent of the Arctic in 2020", because as the authors also mention in the discussion, the high-pressure in July 2020 also contributed to sea ice melt, so in my view, conditions well aligned for setting a new record minimum in July sea ice. This is not reflected in the current last sentence of the Abstract, while the authors acknowledge it in the discussion (line 485-487).

Response: Thanks, we used a new title "Contribution of warm and moist atmospheric flow to a record minimum July sea ice extent of the Arctic in 2020" for this paper as other factors such as the high-pressure in July 2020 contributed to the minimum SIE as well. Besides, in the last sentence of the Abstract, the role of the anticyclonic atmospheric circulation pattern has been reflected by the expression "large-scale atmospheric circulation" (low-pressure and high pressure).

Furthermore, the low-pressure anomaly in the central Arctic is insignificant compared with the climatology and the high-pressure anomaly over Siberia is more anomalous, thus might have played a larger role than the low pressure in the central Arctic in setting the stage for July extreme melt; in light of the composites, it will be easier to decide. In lines 176-180 please clarify, which centers are statistically significant, not all of those that are currently mentioned in the text. In the discussion also, line 461 needs to be refined, probably in accordance with the suggested composite analysis.

Response: Indeed, the low-pressure anomaly in the central Arctic exceeds one standard deviation, while the high-pressure anomaly over Siberia and the low-pressure region in the Barents Sea have magnitudes larger than 1.5 standard deviations. Based on the analysis, the low-pressure center in

the central Arctic, albeit with a smaller magnitude, also plays an important role in the sea ice melt in the study region. That is to say, it acted together with the high-pressure anomaly over Siberia to generate strong winds in the study area. These winds divert the moisture and energy fluxes to prevail in the Laptev and East Siberian Seas after they entered the Arctic. To show the magnitude of the low-pressure anomaly in the central Arctic, we have revised show the values where anomalies exceed one standard deviation in Fig.2 in the manuscript. However, in the difference map (Fig. 1 in the response letter), the high-pressure anomaly over Siberia is not discernible and the low-pressure center near the pole is not conspicuous, which again highlighted the unique atmospheric condition in spring 2020. The related description has been clarified in the revised manuscript (Line 188-192, 200 in the revised manuscript).

R.2. I am unsure what the authors mean by 'average melt days' in the caption of Fig. 6. Based on the text, it should refer to that the authors averaged the melt onset days in all grid points corresponding to the study area in a given year. If so, please clarify the figure caption, e.g., instead of 'the averaged melt date of the study area' say 'day of the year corresponding to the melt onset in each year averaged over the study area'. Is the melt onset in 2020 outside 1.5 standard deviation from the mean of all melt onset dates?

Response: Yes, Fig. 6b shows the melt onset date in each year averaged over the study area during the period 1979-2020. The caption of Fig. 6 has been accordingly. Yes, the melt onset in 2020 lies outside 1.5 standard deviations from the mean averaged over the study area.

**Minor concerns**

R.3. Figure 3 lacks the hatching indicating statistically significant grid points.

Response: Modified accordingly, we have added dots to denote anomalies that exceed two standard deviations (Fig. 3 in the revised manuscript).

R.4. The authors may consider replotting their figures with using a more commonly used projection with central longitude 0° instead of the currently used ones centered over Greenland. I think this would make the figures easier to read and compare with other papers on similar topics.

**Response: Modified as suggested.**

**Response to Referee #3**

**Summary:**

Liang et al. examine the processes that may have contributed to the extreme Arctic sea ice loss event in the summer of 2020. Analysis using ERA5 and PIOMAS products shows enhanced poleward moisture and energy transport likely generated increased downwelling surface radiative fluxes that melted the ice. The study then links this meridional transport to increased cyclone activity and intensity during spring of that year.

**General comments:**

When considering preconditioning of the ice in spring prior to the melt season, several studies have made linkages to poleward energy and moisture transport, but it is interesting to see this applied to 2020, which may be an important year for understanding extreme sea ice loss events. The discussion that places 2020 in the context of the last 4 decades, for example in Figure 6 and 10, is particularly interesting. The goals, methods, and logic are overall valid and suitable. The presentation of the results and conclusions could use some minor revision to improve the clarity and to help tie everything together.

It seems that good improvements were made in the previous round of revisions. Therefore, I only have recommendations for minor revisions detailed below. These comments are mainly about clarifying certain aspects of the figures or changing the figures to make it easier to follow the connections between them. The main comment is that the interpretation of Figure 4 may be overstated and the counterargument using evaporation might not be sufficient since much of the strongest warming and moistening may also occur over land or at the land-sea interface.

Response: Thank you for your general comment. We believe that the interpretation of Figure 4 is not overstated. The same vertical profile as Fig. 4 in the manuscript but excluding the land grids by using an ocean mask is shown in the figure below. Note that the latitude 65-66° N has no ocean grids thus were labeled as no data. Firstly, significantly elevated temperature and higher moisture content distributed widely from 67° N to 80° N over the ocean and extend from the surface into the mid-troposphere. That is to say, even though strong warming and moistening also occur over land or coast, there exist unusual conditions with higher moisture content and warming within the Arctic

atmospheric column over the ice cover region. Then we clarified the source of this extra water vapor (local evaporation or transport from lower latitudes). The below normal evaporation indicates that the enhanced moisture contributing to the moister atmosphere is primarily provided by atmospheric transport from remote areas. Secondly, warming and moistening over land could also contribute to ice melting through several mechanisms, especially for the heat input by river runoff. We have added some discussions on this issue and more explanations about Fig. 4 in Section 3 (Line 221-228 in the revised manuscript).

Figure 2. Same as Fig.4 in the manuscript, with data retrieved over oceanic grids.

**Specific comments:**

Line 210-211: Is this reference for Figure 4?

Response: No, this sentence mentioned the great convergence of energy and moisture, which refers to Fig. 3c and d.

Lines 213-222: The centers of the strongest warming and moistening in Figure 4 not only coincided with the regions of strong sea ice loss but over land, too. Many of the fluxes over land in Figure 5 are opposite to those over the ocean and their effect on the temperature and specific humidity

profiles may be different as well. Are there substantial changes in the anomaly patterns after detrending?

Response: As stated above, strong warming and moistening events occurred over both land and the ocean. They could contribute to the sea ice loss in the study area in July 2020. The intrusion of moisture and energy leads to warming and damping of the atmosphere in the spring months hence changing its radiative characteristics. The net longwave radiation and turbulent fluxes show opposite signs over some parts of the land and the ocean. Yet, enhanced surface fluxes occurred in the study area with considerable ice loss, which is the focus of this study. Additionally, there are no substantial changes in the anomaly patterns after detrending, see Fig. 3 below. The spatial distributions of the anomalies agree well with those related to climatology with little change in the magnitudes.

---

## Author Response (AR3)

**Response to Editor**

Dear Editor David Schroeder:

Thank you very much for handling and review of our manuscript. We have read the technical comments made by Referee #3, which help us improve our research. In the latest revised manuscript, we have corrected the typos and referred to the recommended literature when discussing the radiative effects of cloud on reanalysis and the (Line 269-275) and the driving mechanisms of early melt onset (Line 300-306). Note that we have summarized the existing knowledge about the effects of enhanced moisture and energy transport on the melt onset of the sea ice surface and reorganized these sentences for a better expression (Line 300-306).

Best regards

Yu Liang and co-authors